# Private Testing of Distributions via Sample Permutations

**Maryam Aliakbarpour**
CSAIL, MIT
maryama@mit.edu

**Ilias Diakonikolas**
University of Wisconsin, Madison
ilias.diakonikolas@gmail.com

**Daniel Kane**
University of California, San Diego
dakane@ucsd.edu

**Ronitt Rubinfeld**
CSAIL, MIT, TAU
ronitt@csail.mit.edu

## Abstract

Statistical tests are at the heart of many scientific tasks. To validate their hypotheses, researchers in medical and social sciences use individuals' data. The sensitivity of participants' data requires the design of statistical tests that ensure the privacy of the individuals in the most efficient way. In this paper, we use the framework of property testing to design algorithms to test the properties of the distribution that the data is drawn from with respect to differential privacy. In particular, we investigate testing two fundamental properties of distributions: (1) testing the equivalence of two distributions when we have unequal numbers of samples from the two distributions. (2) Testing independence of two random variables. In both cases, we show that our testers achieve near optimal sample complexity (up to logarithmic factors). Moreover, our dependence on the privacy parameter is an additive term, which indicates that differential privacy can be obtained in most regimes of parameters for free.

## 1 Introduction

We study questions in *statistical hypothesis testing*, a field of statistics with fundamental importance in scientific discovery. At a high level, given samples from an unknown statistical model, the goal of a hypothesis test is to determine whether the model has a desired property. The first — and arguably the most fundamental — objective in hypothesis testing is to make an accurate determination *with as few samples as possible.* In this work, we focus on understanding the trade off between sample size and an additional important criterion — preserving the *privacy* of the underlying data sets.

Early work in statistics [Pea00, NP33] studied the asymptotic regime, where the sample size goes to infinity. In this paper, we are interested in obtaining finite sample bounds in the *minimax setting* that has been extensively studied in the computer science and information theory communities during the past couple of decades. More specifically, we will work with the formalism of *distribution property testing* [BFR+00, BFR+13]: Given samples from a collection of unknown probability distributions over discrete domains, do the underlying distributions satisfy a desired property $\mathcal{P}$ or are they "far" from satisfying the property? (The definition of "far" is typically quantified via some global error metric, e.g., the total variation distance. See Preliminaries section.) The goal is to develop testers for various properties with information-theoretically optimal sample complexity, which is typically *sublinear* in the domain sizes of the underlying distributions. We note that, in recent years, such sample-optimal methods have been obtained for testing a range of properties, including identity testing ("goodness-of-fit"), closeness testing ("equivalence testing" or "two-sample testing"), and independence testing.

The primary goal in classical statistics theory is to minimize the sample size of inference tasks. In recent years, a wide range of settings involves performing hypothesis testing tasks on sensitive data representing specific individuals, such as data describing medical or other behavioral phenomena. In such cases, the outputs of standard tests may reveal private information that should not be divulged. Differential privacy [Dwo09, DR14a] is a formal framework (see Preliminaries section) that may allow us to obtain the scientific benefit of statistical tests without compromising the privacy of the underlying individuals. Intuitively, differential privacy postulates that similar data sets should have statistically close outputs; and that once this guarantee is achieved, then provable privacy is preserved. Differentially private data analysis is a very active research area, in which a wealth of techniques have been developed for a range of tasks.

When designing a differentially private hypothesis testing algorithm, there are two criteria to balance. On the one hand, we require that the algorithm satisfies the differential privacy condition on *any* input dataset. On the other hand, we require that the algorithm is a valid statistical hypothesis tester (i.e., it correctly distinguishes inputs satisfying property $\mathcal{P}$ from inputs that are far from satisfying $\mathcal{P}$). These competing criteria suggest that, in general, the sample size required to ensure both of them grows compared to the non-private setting. Recent work [CDK17, ADR18, ASZ18] has shown that for basic tasks, such as identity testing and equivalence testing, the sample size increase of a differentially private tester compared to its non-private analogue is negligible.

In this work, we continue this line of investigation. We give a new general technique that yields sample-efficient differentially private testers and apply it for two fundamental statistical tasks: the problem of equivalence testing *with unequal sized samples* and the problem of *independence testing* (defined in the following paragraph). Notably, prior techniques were inherently unable to provide sample-efficient private testers for either of these problems.

**Our Contributions.**    The main contribution of this work is a general technique for preserving privacy that in particular can be used to obtain sample-efficient and differentially private hypothesis testers based on the algorithmic technique in the work of [DK16]. This technique can be applied to several testing problems and in particular, yields the only known sample optimal testers for the following testing problems:

*Equivalence Testing with Unequal Sized Samples:* Given a target error parameter $\epsilon > 0$, $s_1$ independent draws from an unknown distribution $p$ over $[n]$, and $s_2$ draws from an unknown distribution $q$ over $[n]$, distinguish the case that $p = q$ from the case that $\|p - q\|_1 \geq \epsilon$.

*Independence Testing:* Given a target error parameter $\epsilon > 0$, and $s$ independent draws from an unknown distribution $p$ over $[n] \times [m]$, distinguish the case that $p$ is a product distribution (i.e., its two coordinates are independent) versus $\epsilon$-far, in $\ell_1$-distance, from any product distribution.

These problems have been extensively investigated in distribution testing during the past decade [BFF+01, LRR11, CDVV14, VV14, AJOS14, ADK15, BV15, DK16] and sample-optimal testers are known for them in the non-private setting [DK16]. In this work, we design the first differentially private testers for these problems with optimal (or near-optimal) sample complexity. In particular, we show that the sample complexity of both these problems in the private setting is nearly the same as in the non-private setting, i.e., privacy comes essentially for free.

In this work, we focus on privatizing the optimal non-private testers for the above problems presented in [DK16]. The algorithmic technique of [DK16] splits samples into two groups — "flattening samples" and "testing samples" — that are used in very different ways. To obtain privacy guarantees, we must design algorithms that have low sensitivity to the samples — that is, changing one sample does not have much effect on the outcome. Previous works in private testers for hypothesis testing are based on algorithms that use the samples analogously to the use of the "testing samples" in [DK16]. One can similarly use those techniques to design algorithms with low sensitivity with respect to the "testing samples" in our setting. However, since using "flattening samples" is a key to achieve testers with the optimal sample complexity, we need to obtain low sensitivity with respect to the "flattening samples". As the output of the testing algorithm can be very sensitive to small changes in the set of "flattening samples", more care is required in designing low sensitivity algorithms for using the "flattening samples", In a nutshell, what we present is a technique for designing private testers which considers the output of the algorithm in [DK16] on *every* permutation of the samples and outputs a result that is based on the aggregate. In order to show that this approach gives the correct answer, we have to show that not only the expectation of the result is the same as that of

the non-private tester (which is straightforward), but also the variance is small enough so that the resulting output is correct with high probability (which does not follow from the calculation of the variance of the non-private testers). For the independence testing problem, we need an extra step to reduce sensitivity further. We first show that trying every permutation results in low sensitivity in the typical case (over the random samples), and thus gives a private tester. In the non-typical case, more care must be taken — we give an algorithm for modifying the samples in such a way that we can reduce to the typical case. We describe the challenges and techniques in more detail in Section 3. Though our methods are mainly tailored for use with the [DK16] testers, there are several other distribution property testing algorithms which use samples in similar sensitive manners, e.g., [CDGR16]— the hope is that these techniques will be fruitful in allowing those algorithms to be made differentially private as well.

**Related Work**  The field of *distribution property testing* [BFR$^+$00] has been extensively investigated in the past couple of decades, see [Rub12, Can15, Gol17]. A large body of the literature has focused on characterizing the sample size needed to test properties of arbitrary discrete distributions. This regime is fairly well understood: for many properties of interest there exist sample-efficient testers [Pan08, CDVV14, VV14, DKN15b, ADK15, CDGR16, DK16, DGPP16, CDS17, Gol17, DGPP17, BC17, DKS18, CDKS17b]. More recently, an emerging body of work has concenterated on leveraging *a priori* structure of the underlying distributions to obtain significantly improved sample complexities [BKR04, DDS$^+$13, DKN15b, DKN15a, CDKS17a, DP17, DKN17, DKP19].

Differential privacy was first introduced in [DMNS06]. Recently, a new line of research studies distribution testing and learning problems with respect to differential privacy [DHS15, CDK17, ADR18, ASZ18]. The focus of these works is on testing identity and closeness of distributions, leaving other properties mostly unexplored. Other models for distribution testing problems with respect to differential privacy have been studied [WLK15, GLRV16, KR17, KFS17]. In most of these latter works, only a type I error guarantee is provided, which is a significantly weaker guarantee compare to ours. In addition, other settings for privacy, e.g, local privacy, is investigated [She18, GR18, ACFT19].

## 2 Preliminaries

### 2.1 Definitions

**Notation:**  We use $[n]$ to denote the set $\{1, 2, \ldots, n\}$. We consider discrete distributions over a finite domain, in particular, over $[n]$ without loss of generality. For a distribution $p$, we write $p(i)$ to denote the probability of element $i$ in $[n]$. One can assume each distribution has an associated probability function $p : [n] \rightarrow [0, 1]$ such that $p(i)$'s are non-negative and $\sum_{i \in [n]} p(i) = 1$. For set $S \subseteq [n]$, $p(S)$ denotes the total probability of the elements in $S$ (i.e., $\sum_{i \in S} p(i)$). Note that one can think of each discrete distribution over $[n]$ as a vector in $\mathbb{R}^n$ where the $i$-th coordinate is the probability of element $i$. Having said that, we can define the $\ell_k$-norm of a distribution in the same manner as of a vector: For a vector $x \in \mathbb{R}^n$ and any $k > 0$, the $\ell_k$-norm of $x$ is equal to $\left( \sum_{i \in [n]} |x_i|^k \right)^{\frac{1}{k}}$, and is denoted by $\|x\|_k$. In addition, the $\ell_k$-distance between two distributions $p$ and $q$ over $[n]$ is equal to $\|p - q\|_k$. Throughout this paper, we use the $\ell_1$-distance to measure the discrepancy between distributions, which is equivalent to the total variation distance up to a factor of two. In particular, we say distribution $p$ is $\epsilon$-far from distribution $q$ if $\|p - q\|_1 \geq \epsilon$. We use $\mathbf{Lap}(\lambda)$ to denote the zero mean Laplace distribution with parameter $\lambda$. The probability density function of the Laplace distribution with parameter $\lambda$ at point $x \in \mathbb{R}$ is $\mathbf{Lap}(x; \lambda) = e^{e^{-|x|/b}}/(2b)$. We use the notation for the distribution and a random variable exchangeably where the difference is clear from the context.

**Distribution Testing:**  Formally, we define a property $\mathcal{P}$ to be a set of distributions. We say a distribution $p$ has the property $\mathcal{P}$ if $p \in \mathcal{P}$; and, we say $p$ is $\epsilon$-far from having the property $\mathcal{P}$ when $p$ is $\epsilon$-far from all distributions in $\mathcal{P}$. Assume an algorithm has sample access to a distribution $p$. We say the algorithm is an $(\epsilon, \delta)$-*tester for property* $\mathcal{P}$, if the following holds with probability at least $1 - \delta$:

- **Completeness case:** If $p$ has the property $\mathcal{P}$, then the algorithm outputs accept.
- **Soundness case:** If $p$ is $\epsilon$-far from $\mathcal{P}$, then the algorithm outputs reject.

Analogously, one can generalize the above definition to the case that we have sample access to two distributions. In particular, given sample access to two distributions $p$ and $q$ over $[n]$, an $(\epsilon, \delta)$-tester for *testing closeness* of $p$ and $q$ distinguishes the following cases with probability at least $1 - \delta$:

- **Completeness case:** If $p$ is equal to $q$, then the algorithm outputs accept.
- **Soundness case:** If $p$ is $\epsilon$-far from $q$, then the algorithm outputs reject.

**Testing closeness of distributions via flattening technique:**    In this paper, we build on the non-private closeness tester presented in [DK16, CDVV14]. Here, we give an overview of the closeness tester, and the flattening technique which turn it into a tester with the optimal sample complexity.

Suppose we have sample access to two distributions $p$ and $q$ on the domain $[n]$. Let $s$ be a parameter that determines the expected number of samples. We draw $\mathbf{Poi}(s)$ samples from $p$ and $q$. For each $i \in [n]$, let $X_i$ and $Y_i$ denote the number of occurrences of element $i$ in the sample sets from $p$ and $q$ respectively. In [CDVV14], the authors proposed the following statistic to test the closeness of $p$ and $q$: $Z = \sum_{i=1}^{n}(X_i - Y_i)^2 - X_i - Y_i$. The expected value of $Z$ is proportional to the squared $\ell_2$-distance of $p$ and $q$ which enables us to use $Z$ for testing closeness of $p$ and $q$. While this statistic is mainly measuring the $\ell_2$-distance, one can use it for testing in $\ell_1$-distance as well by using trivial inequalities between the distances. By careful analysis of the variance of $Z$, it is shown that $Z$ concentrates around its expectation when we draw at least $s = \Theta(n/(\epsilon^2 \max(\|p\|_2, \|q\|_2)))$ samples[1]. More specifically, it is shown that in the case where $p = q$, $Z$ is below a threshold parameter, $\tau$; and when $p$ is $\epsilon$-far from $q$, $Z$ is at least $\tau$ with high probability. Thus, we can test the closeness of $p$ and $q$ by computing $Z$ from a large enough sample set and comparing it with the threshold $\tau$.

The above algorithm would be sample-efficient only when $\max(\|p\|_2, \|q\|_2)$ is not too large. In [DK16], the authors provide a technique, called *flattening*, that decreases the $\ell_2$-norm of a distribution. Using this technique, they map $p$ and $q$ to two other distributions at least one of which has smaller $\ell_2$-norm. Then, they obtain a sample-optimal $\ell_1$-closeness tester for $p$ and $q$ by showing its equivalence to closeness testers for the two distributions obtained after flattening.

We discuss the flattening technique in more detail. To flatten a distribution $p$, we need a (multi)set of the domain elements denoted by $F$. This set is usually obtained by drawing samples from the underlying distributions, and the elements in $F$ are called *flattening samples*. Using flattening samples, we transform $p$ to another distribution $p^{(F)}$ over a larger domain in a way that the $\ell_2$-norm of $p^{(F)}$ is small. We build the new domain for $p^{(F)}$ as follows: For each element $i$ in the domain of $p$, we first count the number of occurrences of $i$ in $F$, namely $k_i$, and put $b_i := k_i + 1$ elements associated to $i$ in the new domain. We refer to the elements of the new domain as *buckets*. We define $p^{(F)}$ to be the distribution that assigns the probability mass of $p(i)/b_i$ to all the $b_i$ buckets of $i$ for all $i$ in the domain. Note that one can generate a sample from $p^{(F)}$ upon receiving a sample from $p$: For a fresh sample $X = i$ drawn from $p$, the flattening procedure maps it to a randomly selected bucket $j$ among the $b_i$ buckets of $i$. Then, it outputs $X' = (i, j)$ as a sample from $p^{(F)}$. The above procedure has several important properties which help us later in our analysis:

- By the above construction, it is clear that the size of the new domain is $\sum_i b_i = n + |F|$. Thus, as long as $|F|$ is not larger than $n$ (in the regime where we have sublinear number of samples), the size of domain increases only by a constant factor.

- It is shown that if $F$ contains $\mathbf{Poi}(k)$ samples from $p$, then the expected $\ell_2$-norm of $p^{(F)}$ is at most $1/k$.

- If we flatten two distributions using the same assignments for the buckets (i.e., the flattening set $F$), the $\ell_1$-distance between the two distributions remains unchanged. Thus, it suffices to test closeness of $p^{(F)}$ and $q^{(F)}$ for testing the closeness of $p$ and $q$.

**Privacy:**    We say two sample sets, $X$ and $X'$, from a universe $[n]$ are *neighboring* if and only if their Hamming distance is one (meaning that they differ in exactly *one* sample). A randomized algorithm $\mathcal{A}$ is $\xi$-private if for any subset $S$ of the possible outputs of the algorithm, {accept, reject} in the context of this paper, and for any two neighboring $X$ and $X'$, the following holds:

$$\mathbf{Pr}[\mathcal{A}(X) \in S] \leq e^\xi \cdot \mathbf{Pr}[\mathcal{A}(X') \in S].$$

For a function $f$ over sample sets, the *sensitivity of $f$* is defined as follows:

$$\Delta(f) = \max_{X,X'} |f(X) - f(X')|,$$

where the maximum is taken over all possible sample sets that differ in only one sample. A standard method for making functions private is the *Laplace mechanism* [DR14b]. In this mechanism, to make a function $f$ $\xi$-differentially private, one adds Laplace noise to the output of function: $\hat{f}(X) := f(X) + \mathbf{Lap}(\Delta(f)/\xi)$. It is easy to show that $\hat{f}$ satisfies the definition of differential privacy with parameter $\xi$.

# 3   General approach for making closeness-based testers private

As we mentioned earlier, several properties can be tested via a reduction to the flattening-based closeness tester. These reductions and the resulting optimal testers were presented in [DK16]. In this section, we focus on describing a general approach for making such testers differentially private. We start by explaining the structure of the existing reductions in the non-private setting. Next, we explain our main idea for making the reductions and the testers private, which is to derandomize the non-private tester. Then we give the characteristics of the reductions that can be turned into a private algorithm (see Definition 3.2). In particular, if *any* property can be tested via a desired reduction to the closeness testing problem, it can be also privately tested with a reduction to our general private closeness tester. At the end, we describe our algorithm and prove its correctness in the full version.

## 3.1   Reduction procedure in non-private setting

In this section, we elaborate on the structure of the reductions to the closeness tester with the use of the flattening technique proposed in [DK16]. Suppose we aim to test whether a distribution[2] has property $\mathcal{P}$ or it is $\epsilon$-far from any distribution in $\mathcal{P}$ via a reduction to the flattening-based closeness tester in the non-private setting. The reduction has the following structure: Upon receiving a sample set from the underlying distribution, the *reduction procedure* splits the samples into two sets *test samples*, denoted by $T$, and *flattening samples*, denoted by $F$. The reduction procedure uses these sample sets as follows:

- **Test samples**: The reduction procedure use the test samples to generate samples from two distributions $p$ and $q$ over a domain of size $n$. The distributions $p$ and $q$ are designed in such a way that if the underlying distribution has the property $\mathcal{P}$, then $p$ and $q$ are the same; and if the underlying distribution is $\epsilon$-far from any distribution that has the property, then $p$ and $q$ are $\Theta(\epsilon)$-far from each other as well. This transformation is essentially the core of the reduction to the testing closeness problem.

- **Flattening samples**: In addition to samples from $p$ and $q$, the reduction procedure generates $n$ positive integers $b_1, b_2, \ldots, b_n$ that indicate the number of buckets for each domain element. These numbers are used for flattening of $p$ and $q$. (See the Preliminaries section for more details.)

An example of such reductions is testing independence. Suppose $d$ is a distribution over $[n] \times [m]$, and our goal is to test whether the two coordinates of the samples drawn from $d$ are independent or not. It is not hard to see that this problem is equivalent to testing whether $p := d$ is equal to $q := d_1 \times d_2$, where $d_1$ and $d_2$ are the two marginal distributions of $d$. For more examples, see [DK16].

After the reduction procedure generates samples from $p$ and $q$, and the number of buckets for each domain element, we use the flattening-based closeness tester for the testing closeness of $p$ and $q$. As we explain in the Preliminaries section, first, we transform $p$ and $q$ to $p^{(F)}$ and $q^{(F)}$ on the new domain:

$$D = \{(i,j) | i \in [n] \text{ and } j \in [b_i]\}.$$

The hope is that after flattening the $\ell_2$-norm of at least one of the resulting distributions is small. Recall that the probability of each element $(i,j)$ in $D$, i.e., a bucket, according to $p^{(F)}$ is $p(i)/b_i$.

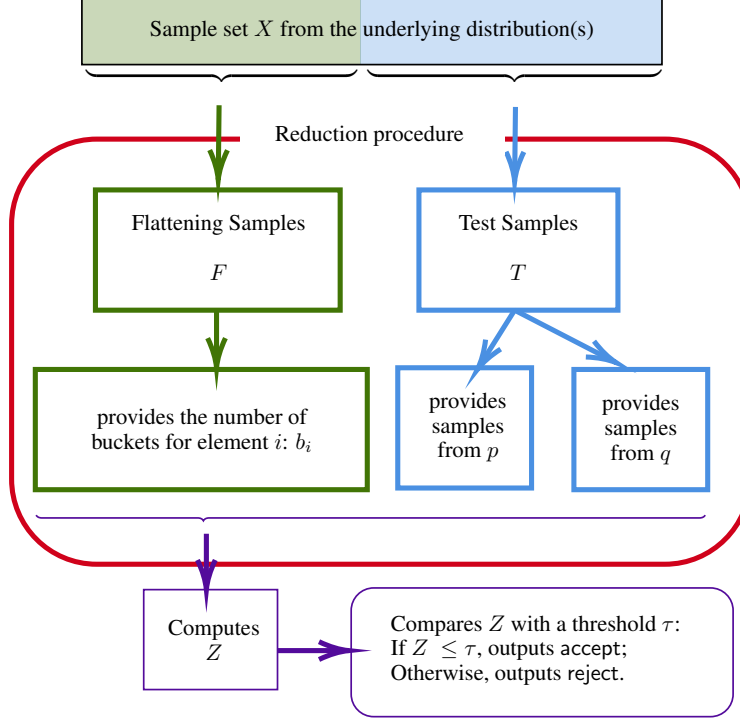

Figure 1: Standard reduction procedure to testing closeness of two distributions.

The closeness tester transforms samples from $p$ to samples from $p^{(F)}$ (and similarly for $q$). For each sample $i$ from $p$, the closeness tester assigns it to the bucket $(i, j)$, where $j$ is picked uniformly at random from $[b_i]$. We denote the number of occurrences of bucket $(i, j)$ in the sample set from $p^{(F)}$ (and $q^{(F)}$) by $v_{i,j,1}$ (and $v_{i,j,2}$). The flattening-based closeness tester computes the following statistic $Z$:

$$Z := \sum_{i=1}^{n} \sum_{j=1}^{b_i} (v_{i,j,1} - v_{i,j,2})^2 - v_{i,j,1} - v_{i,j,2} \ , \tag{1}$$

and compares it with a threshold to establish whether $p^{(F)}$ and $q^{(F)}$ are equal or $\Theta(\epsilon)$-far from each other. Since the transformation to $p^{(F)}$ and $q^{(F)}$ does not change the $\ell_1$-distance between $p$ and $q$, the output of the tester determines whether $d$ has the property $\mathcal{P}$ or $\epsilon$-far from it. See Figure 1 for an overview of this process.

### 3.2 Derandomizing the non-private tester

To develop a differentially private algorithm for closeness testing with flattening, we "derandomize" the standard non-private closeness tester provided in [DK16, CDVV14]. The derandomization of the tester results in a stable statistic, which means if we change one sample in the sample set, the value of the statistics does not change drastically. The stability implies that the statistic has low sensitivity, and it can be privatized using fewer number of extra samples.

In the previous section, we explained how the reduction and the tester work. Note that there are two steps that the algorithm can make random choices: (i) The algorithm splits the samples into two sets $F$ and $T$ (flattening and test samples): Upon receiving a set of $s + k$ samples, $X = \{x_1, x_2, \ldots, x_{s+k}\}$, where $s$ and $k$ is the number of test samples and the number of flattening samples respectively, usually the algorithm assigns the first $s$ samples to the test set, and the rest to the flattening set. Equivalently, we can view this step as follows: the algorithm permutes the sample according to a *random* permutation $\pi$. Then, it splits them into $F$ and $T$. In other words, $T$ is equal to $\{x_{\pi(1)}, x_{\pi(2)}, \ldots, x_{\pi(s)}\}$, and $F$ is equal to $\{x_{\pi(s+1)}, x_{\pi(s+2)}, \ldots, x_{\pi(s+k)}\}$. (ii) The algorithm randomly selects a bucket for each sample from $p$ and $q$. Let $r$ denote the string of random bits that the algorithm uses to choose the buckets. We eliminate the randomness of these two steps

by setting our new statistic, $\overline{Z}$, to be the *expected value* of the statistic $Z$ over the random choices of the algorithm. More precisely, for a given input sample set $X$, we define:

$$\overline{Z}(X) := \mathbf{E}_{\pi,r}[Z|X] \ .$$

We can simplify the above statistic a step further by computing the closed-form of the expected value of $Z$ over all the random choices of $r$, i.e., $\mathbf{E}_r[Z|X,\pi]$. We provide the exact value in the following lemma, and it is proved in the full version.

**Lemma 3.1.** *Let $s_{i,1}$ (similarly $s_{i,2}$) be the number of occurrences of element $i$ in the sample sets of $p$ ($q$). Assume $b_i$ is the number of buckets assigned to element $i$. Let $v_{i,j,1}$ (similarly $v_{i,j,2}$) be the number of occurrences of bucket $(i,j)$ in the sample set from $p^{(F)}$ ($q^{(F)}$). Then, we have:*

$$\mathbf{E}_r\left[\sum_{j=1}^{b_i}(v_{i,j,1}-v_{i,j,2})^2 - v_{i,j,1} - v_{i,j,2}\ \middle|\ b_i, s_{i,1}, s_{i,2}\right] = \frac{(s_{i,1}-s_{i,2})^2 - s_{i,1} - s_{i,2}}{b_i}\ ,$$

*where the expectation is taken over all random assignments of the samples to the buckets.*

The lemma above immediately implies the following equation:

$$\overline{Z}(X) = \mathbf{E}_\pi\left[\mathbf{E}_r[Z|X,\pi]|X\right] = \mathbf{E}_\pi\left[\sum_{i=1}^{n}\frac{(s_{i,1}-s_{i,2})^2 - s_{i,1} - s_{i,2}}{b_i}\ \middle|\ X\right]\ . \tag{2}$$

For the rest of this paper, we work with this latter form of $\overline{Z}$.

### 3.3   Designing a general private tester

Note that the algorithm we want to design has to satisfy two guarantees: First, it should be an accurate tester, i.e., it should output the correct answer with high probability. Second, it should be a differentially private algorithm.

For the accuracy guarantee, we first, need to show that the proposed statistic, $\overline{Z}$, is sufficient for testing closeness of $p$ and $q$, and ultimately testing property $\mathcal{P}$. At first glance, the claim seems trivially true: We know that for *any* sample set $X$ and *any* random choice of $\pi$, and $r$, the statistic $Z$ will be a sufficient statistic for testing property $\mathcal{P}$ just because of the properties of the non-private tester. Since $\overline{Z}$ is essentially the expected value of a group of sufficient statistics, it must be immediate that $\overline{Z}$ is a sufficient statistic as well. However, there is a subtle difference between the guarantees we require for the statistics $Z$ and $\overline{Z}$. To analyze the standard tester in [DK16], the authors showed that with high probability the set of flattening samples decreases the $\ell_2$-norm of one of the two distributions $p$ and $q$. The low $\ell_2$-norm property is sufficient to show that $Z$ has low variance, and one can use it for closeness testing. In fact, the authors show that the statistic $Z$ works for some flattening test which occurs with high constant probability. However, in our setting, we wish to show the variance of $\overline{Z}$ is low over the random choices of both the flattening samples and the test samples. Hence, while we are taking out the randomness of $r$, or $\pi$, we are introducing a new source of randomness, which is the set $F$. Thus, it is unclear whether $\overline{Z}$ can be used as a statistic for the problem or not.

The goal for the rest of this section is to prove that if the reduction procedure has the desired characteristics, then $\overline{Z}$ is a sufficient statistic for testing property $\mathcal{P}$. We say a reduction procedure, $\mathcal{A}$, is a *proper procedure* if it has these desired guarantees, which we formally define in the following definition:

**Definition 3.2** (Proper procedure). *Let $\mathcal{A}$ be a procedure that reduces testing property $\mathcal{P}$ to testing closeness of two distributions $p$ and $q$ over $[n]$ given an input sample set $X$. We say $\mathcal{A}$ is a* proper *procedure if $\mathcal{A}$ flattens $p$ and $q$ in such a way that the following holds for two non-negative constants $c_0 < 1$, and $c_1 \geq 1$:*

$$\mathbf{Pr}_X\left[\mathbf{E}_\pi\left[\|p^{(F)}-q^{(F)}\|_2^2\ \middle|\ X,\pi\right] \geq 4c_0 \cdot \mathbf{E}_F\left[\|p^{(F)}-q^{(F)}\|_2^2\right]\right] \geq 0.9\ , \tag{3}$$

$$\mathbf{E}_F\left[\|p^{(F)}-q^{(F)}\|_4^4\right] \leq c_1 \cdot \left(\mathbf{E}_F\left[\|p^{(F)}-q^{(F)}\|_2^2\right]^2\right)\ . \tag{4}$$

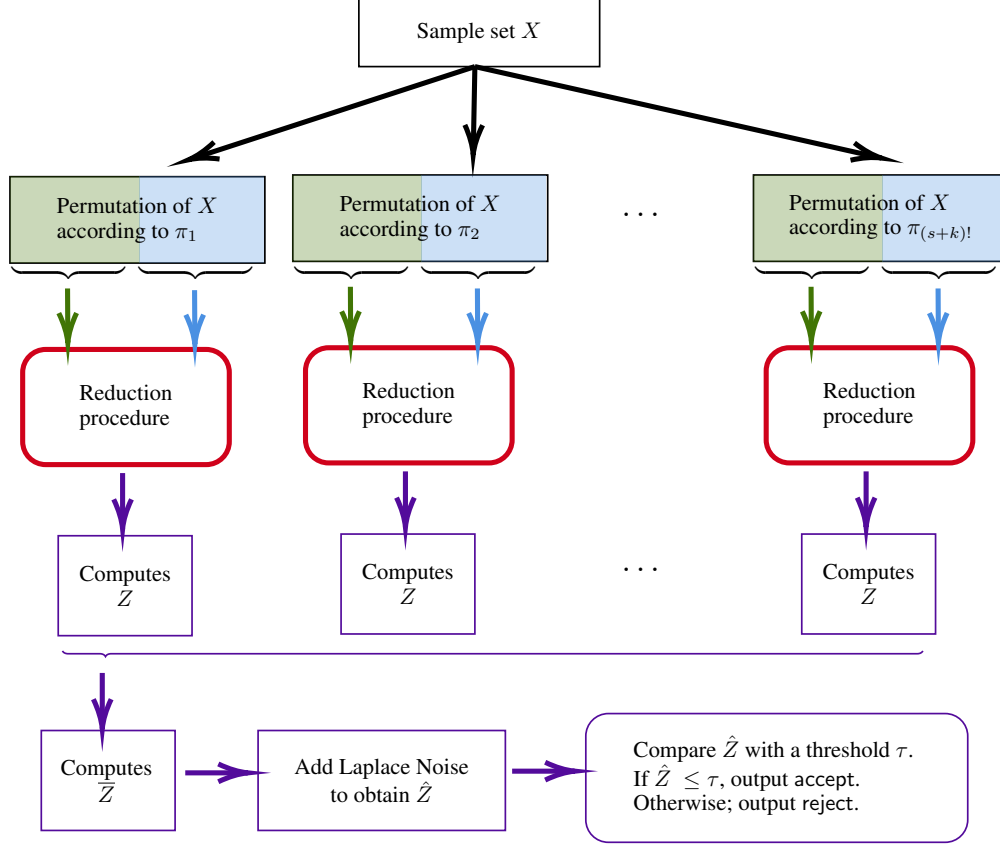

Figure 2: Our approach which uses $\overline{Z}$ instead of $Z$ to reduce sensitivity.

For the privacy guarantee, using the statistic $\overline{Z}$, we design a $\xi$-private algorithm. To make the statistic $\xi$-differentially private, we use a standard technique in differential privacy, the Laplace mechanism: we add $\mathbf{Lap}(\Delta(\overline{Z})/\xi)$ to the statistic, to make it $\xi$-differentially private. Note that $\Delta(\overline{Z})$ is the sensitivity of $\overline{Z}$. The importance of the idea of taking $\overline{Z}$ instead of $Z$ as the statistic is that it results in a stable statistic with low sensitivity. Thus, the magnitude of the noise we add is small, and we can achieve nearly optimal sample complexity. Note that the exact value of $\Delta(\overline{Z})$ depends on the reduction procedure. We bound this quantity for each of the properties we consider separately. However, in this section, we state our result in terms of $\Delta(\overline{Z})$.

Finally, we propose Algorithm 1 for differentially private testing of property $\mathcal{P}$. We analyze the correctness of the algorithm in Theorem 3.3. We provide an illustration of our approach in Figure 2 in comparison with the standard approach in Figure 1.

**Theorem 3.3.** *Let $\mathcal{A}$ be a proper procedure for testing property $\mathcal{P}$ as defined in Definition 3.2. Suppose the expected number of test samples, $s$, is bounded from below:*

$$s \geq \Theta \left( \frac{n \cdot \sqrt{\mathbf{E}_F\big[\min\big(\|p^{(F)}\|_2^2, \|q^{(F)}\|_2^2\big)\big]}}{\epsilon^2} + \frac{\sqrt{n\Delta(\overline{Z})}}{\epsilon\sqrt{\xi}} \right) .$$

*Then Algorithm 1 is a $\xi$-differentially private $(\epsilon, 3/4)$-tester for testing property $\mathcal{P}$.*

**Remark 3.4** (Sample complexity of the algorithm). Our algorithm receives two parameter $s$ and $k$ for the number of test and flattening samples. Our analysis is based on the Poissonization method, so the algorithm is required to generate $\hat{s} := \mathbf{Poi}(s)$ samples from $p$ and $q$, and $\hat{k} := \mathbf{Poi}(k)$ samples for flattening. In this section, for simplicity, we use $s$ and $k$ as the number of samples. Note that we assume that one can generate a sample from $p$ and $q$ using $\Theta(1)$ samples from the underlying distribution, and with probability $0.99$, $\hat{s}$ and $\hat{k}$ are within a constant factor of their expectation. Thus, the sample complexity remains $\Theta(s + k)$.

---

**Algorithm 1** A private procedure for property testing

---

1: **procedure** PRIVATE TESTER$(n, \epsilon, s, k)$
2:     $X \leftarrow$ Draw $s + k$ samples, $x_1, x_2, \ldots, x_{s+k}$ from the underlying distribution.
3:     $\overline{Z} \leftarrow 0$
4:     **for** each permutation $\pi$ **do**
5:         $T \leftarrow x_{\pi(1)}, \ldots, x_{\pi(s)}$
6:         $F \leftarrow x_{\pi(s+1)}, \ldots, x_{\pi(s+k)}$
7:         $S_p \leftarrow \mathcal{A}$ determines a set of samples from $p$ using the test samples $T$
8:         $S_q \leftarrow \mathcal{A}$ determines a set of samples from $q$ using the test samples $T$
9:         **for** $i = 1, 2, \ldots, n$ **do**
10:             $b_i \leftarrow \mathcal{A}$ determines the number of buckets for element $i$ using flattening samples $F$.
11:             $s_{i,1} \leftarrow$ Number of occurrences of element $i$ in $S_p$
12:             $s_{i,2} \leftarrow$ Number of occurrences of element $i$ in $S_q$
13:             $\overline{Z} \leftarrow \overline{Z} + \frac{(s_{i,1} - s_{i,2})^2 - s_{i,1} - s_{i,2}}{b_i} \cdot \mathbf{Pr}[\text{probability of picking } \pi]$
14:     $n \leftarrow \mathcal{A}$ determines an upper bound for the new domain size.
15:     $\hat{Z} \leftarrow \overline{Z} + \mathbf{Lap}(\Delta(\overline{Z})/\xi)$
16:     **if** $\hat{Z} \leq \tau$ **then**
17:         Output accept.
18:     **else**
19:         Output reject.

---

**Remark 3.5.** Although the running time of Algorithm 1 is exponential as stated, one can run it in Poly$(s)$ time as follows: For each domain element $i$ and three numbers $a$ and $b$, and $c$, one can calculate the probability of $S_{i,1} = a, s_{i,2} = b, b_i = c$, so once can compute $\mathbf{E}_\pi\big[(s_{i,1} - s_{i,2})^2 - s_{i,1} - s_{i,2})/b_i\big]$, and $\overline{Z}$ without trying all permutations $\pi$.

### 3.4 Applications of Our Framework

We use our general private tester to achieve differentially private algorithms for the two distribution testing problems we mentioned earlier: (i) Testing closeness of two distributions with unequal sized sample sets, (ii) Testing independence. As stated earlier, our approach is to use the non-private testers for these problems and show they satisfy the proper procedure definition (Definition 3.2). Then, using our general methodology, we achieve a near sample-optimal private testers for these problems. In particular, we have the following theorems. For more information, see the full version.

**Theorem 3.6.** *Suppose $p$ and $q$ are two distributions over $[n]$. There exists a $\xi$-differentially private $(\epsilon, 2/3)$-tester for closeness of $p$ and $q$ that uses $k_1 = \Omega(\max(n^{2/3}/\epsilon^{4/3}, \sqrt{n}/\epsilon\sqrt{\xi}))$ samples from $p$, $\Theta(\max(n/(\epsilon^2\sqrt{\min(n, k_1)}), \sqrt{n}/\epsilon^2, \sqrt{n}/\epsilon\sqrt{\xi}, 1/\epsilon^2\xi))$ from both $p$ and $q$.*

**Theorem 3.7.** *Let $p$ be a distribution over $[n] \times [m]$. There exists a $\xi$-differentially private $(\epsilon, 2/3)$-tester for the testing independence of $p$ that uses $\Theta(s)$ samples where $s$ is:*

$$s = \Theta\left(\frac{n^{2/3}m^{1/3}}{\epsilon^{4/3}} + \frac{(m\,n)^{1/2}}{\epsilon^2} + \frac{(m\,n\log n)^{1/2}}{\epsilon\sqrt{\xi}} + \frac{\log n}{\epsilon^2\xi}\right).$$

### Acknowledgments

MA is supported by funds from the MIT-IBM Watson AI Lab (Agreement No. W1771646), the NSF grants IIS-1741137, and CCF-1733808. ID is supported by NSF Award CCF-1652862 (CAREER), NSF AiTF award CCF-1733796 and a Sloan Research Fellowship. DK is supported by NSF Award CCF-1553288 (CAREER) and a Sloan Research Fellowship. RR is supported by funds from the MIT-IBM Watson AI Lab (Agreement No. W1771646) the NSF grants CCF-1650733, CCF-1733808, IIS-1741137 and CCF-1740751.

## Footnotes

[1]In fact, as a byproduct of our analysis, one can show it suffices to have $s \geq \Theta(n/(\epsilon^2 \min(\|p\|_2, \|q\|_2)))$.

[2]We may have more than one underlying distributions.

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
