[Supplementary Material]

## A  Proof of Theorem 3.3

**Theorem 3.3.** *Let $\mathcal{A}$ be a proper procedure for testing property $\mathcal{P}$ as defined in Definition 3.2. Suppose the expected number of test samples, $s$, is bounded from below:*

$$s \geq \Theta\left(\frac{n' \cdot \sqrt{\mathbf{E}_F\big[\min\big(\|p^{(F)}\|_2^2, \|q^{(F)}\|_2^2\big)\big]}}{\epsilon^2} + \frac{\sqrt{n'\Delta(\overline{Z})}}{\epsilon\sqrt{\xi}}\right).$$

*Then Algorithm 1 is $\xi$-differentially private $(\epsilon, 3/4)$-tester for testing property $\mathcal{P}$.*

**Proof:** First, we prove that the algorithm is $\xi$-differentially private. Note that the statistic we use is $\hat{Z}$ which is equal to $\overline{Z}$, as defined in Equation 2, plus a Laplace random variable with mean $\Delta(\overline{Z})/\xi$. According to the Laplace mechanism $\hat{Z}$ is a $\xi$-differentially private quantity, so the output of the algorithm is private.

Now, we prove that the algorithm is an $(\epsilon, 3/4)$ tester as well. At a high level, the expected value of $\overline{Z}$ is zero when $p = q$; whereas it is larger than $\Theta(\tau)$ when $p$ and $q$ are $\epsilon$-far from each other. By analyzing the variance of $\overline{Z}$, and using Chebyshev's inequality, we show that $\overline{Z}$ is close to its expectation. More specifically, we prove the following claims:

- Completeness case: If $p$ is equal to $q$, then $\overline{Z}$ at most than $\tau/2$ with high probability.
- Soundness case: If $p$ is $\epsilon$-far from $q$, then $\overline{Z}$ is at least $3\tau/2$ with high probability.

In addition, the magnitude of the of the Laplace noise we add to $\overline{Z}$ is small with high probability. Set $\tau$, the threshold we used in the algorithm, to be $2c_0 s^2\epsilon^2/n'$. Using the CDF of Laplace distribution, we have

$$\mathbf{Pr}\Big[|\hat{Z} - \overline{Z}| \geq \frac{\tau}{2}\Big] \leq \exp\left(\frac{c_0\, s^2\epsilon^2\xi}{n'\Delta(\overline{Z})}\right) \leq 0.01$$

where the last inequality is true if $s$ is bounded from below as follows for a sufficiently large constant $c_2$:

$$s \geq c_2 \cdot \frac{\sqrt{n'\Delta(\overline{Z})}}{\epsilon\sqrt{\xi}}.$$

If $|\hat{Z} - \overline{Z}|$ is less than $\tau/2$, then our claim above is sufficient to prove that the algorithm is $(\epsilon, 3/4)$-tester. In the completeness case, with high probability, we will have $\hat{Z} < \overline{Z} + \tau/2 \leq \tau$, and in the soundness case, with high probability, we will have $\hat{Z} > \overline{Z} - \tau/2 \geq \tau$. Thus, in both case, the $\hat{Z}$ will be on the correct side of the threshold. Hence, for the rest of the proof, we focus on the proof of the two claims we state.

To show the bounds for $\overline{Z}$, we introduce an auxiliary random variable $W$. We analyze the expected value, and the variance of $W$, and show that with high probability $W$ is around its expectation by Chebyshev's inequality. Then, we use this fact about $W$ to prove that $\overline{Z}$ must be around its expected value as well, and achieve the desired bound for $\overline{Z}$ with hight probability.

More specifically, for a given $X$ and a given $\pi$, we define $W$ as:

$$W = Z - s^2 \cdot \|p^{(F)} - q^{(F)}\|_2^2,$$

where $F$ is the set of flattening samples, $x_{\pi(\hat{s}+1)}, x_{\pi(\hat{s}+2)} \ldots, x_{\pi(\hat{s}+\hat{k})}$. We can similarly define $\overline{W}$ as well:

$$\overline{W}(X) \coloneqq \mathbf{E}_{\pi,r}[W|X].$$

We analyze the expected value and the variance of $\overline{W}$. First, we define the following notations, $d_{\max}^{(F)}$ and $d_{\min}^{(F)}$ to indicate the maximum and the minimum of the two quantities $\|p^{(F)}\|_2^2$ and $\|q^{(F)}\|_2^2$ respectively. The expected value and the variance of $Z$, as defined in Equation 1, is given in the proof of Proposition 3.1 [19], if we fix the set of flattening samples $F$:

$$\mathbf{E}_{T,r}[Z|F] = s^2\|p^{(F)} - q^{(F)}\|_2^2, \quad \text{and} \quad \mathbf{Var}_{T,r}[Z|F] \leq 8s^3\sqrt{d_{\max}^{(F)}}\|p^{(F)} - q^{(F)}\|_4^2 + 8s^2d. \quad (6)$$

Using the above equation, we compute the expected value and the variance of $\overline{W}$. Note that since samples in $X$ are i.i.d, the order of the samples can change neither the expected value nor the variance. Thus, by symmetrization, we can fix an order of the samples, namely $\pi_0$. As mentioned before, the first $\hat{s}$ samples in $X$ are the test samples, and we denote them by $T$, and the next $\hat{k}$ samples for the flattening samples and are denoted by $F$. Since $T$ and $F$ are completely separated and independent, by Equation 6, we have:

$$
\begin{aligned}
\mathbf{E}_X\big[\overline{W}\big] &= \mathbf{E}_X[\mathbf{E}_{\pi,r}[W|X]] = \mathbf{E}_X[\mathbf{E}_r[W|X,\pi_0]] \\
&= \mathbf{E}_F\Big[\mathbf{E}_{T,r}\Big[\,Z - s^2 \cdot \|p^{(F)} - q^{(F)}\|_2^2\,\Big|\,F\Big]\Big] = 0
\end{aligned}
\tag{7}
$$

Moreover, given the variance bound in the Equation 6, we obtain the following bound for the variance of $\overline{W}$, we have:

$$
\begin{aligned}
\mathbf{Var}_X\big[\overline{W}\big] &= \mathbf{Var}_X[\mathbf{E}_{\pi,r}[W|X]] = \mathbf{Var}_X\bigg[\sum_\pi \mathbf{E}_r[W|X,\pi]\cdot\mathbf{Pr}[\pi]\bigg] \\
&= \sum_{\pi_1}\sum_{\pi_2}\mathbf{Pr}[\pi_1]\cdot\mathbf{Pr}[\pi_2]\cdot\mathbf{Cov}_X\big(\mathbf{E}_r[W|X,\pi_1],\,\mathbf{E}_r[W|X,\pi_2]\big) \\
&\le \frac{1}{2}\sum_{\pi_1}\sum_{\pi_2}\mathbf{Pr}[\pi_1]\cdot\mathbf{Pr}[\pi_2]\cdot\big(\mathbf{Var}_X[\mathbf{E}_r[W|X,\pi_1]]+\mathbf{Var}_X[\mathbf{E}_r[W|X,\pi_2]]\big) \\
&= \mathbf{Var}_X[\mathbf{E}_r[W|X,\pi_0]] = \mathbf{Var}_{F,T}[\mathbf{E}_r[W|F,T]] \\
&= \mathbf{E}_F\big[\mathbf{Var}_T[\mathbf{E}_r[W|F]]\big] + \mathbf{Var}_F[\mathbf{E}_{T,r}[W|F]] \\
&\le \mathbf{E}_F\bigg[8\,s^3\cdot\sqrt{d_{\max}^{(F)}}\cdot\|p^{(F)}-q^{(F)}\|_4^2 + 8\,s^2\cdot d_{\max}^{(F)}\bigg].
\end{aligned}
$$

Using Chebyshev's inequality, $\overline{W}$ cannot be far from its expectation which is zero. More precisely, given a constant $c_0$, we prove that there exists a sufficiently large constant $c_2$ such that $\mathbf{Pr}_X\Big[|\overline{W}| \ge c_0 \cdot \frac{s^2\epsilon^4}{n}\Big]$ is at most $0.01$ assuming we have:

$$
s \ge c_2 \cdot \frac{n' \cdot \sqrt{\mathbf{E}_F\Big[d_{\min}^{(F)}\Big]}}{\epsilon^2}\,.
\tag{8}
$$

We consider the soundness case and the completeness case below separately.

**Completeness Case:** If $p$ is equal to $q$, no matter what $F$ and the bucketing are, $\|p^{(F)}-q^{(F)}\|_2^2$ is zero. Therefore, $\overline{W}$ is always equal to $\overline{Z}$ just by definition of $W$. In fact, we have $\overline{Z} = \overline{W} = \overline{W} - \mathbf{E}_X\big[\overline{W}\big]$. Also, the $\ell_2$-norms of $p^{(F)}$ and $q^{(F)}$ are the same. Thus, the minimum and the maximum of $\|p^{(F)}\|_2^2$ and $\|q^{(F)}\|_2^2$ are equal. Thus, the probability of $\overline{Z}$ be above $\tau/2$ is bounded by $0.01$ using the Chebyshev inequality :

$$
\begin{aligned}
\mathbf{Pr}_X\Big[\overline{Z}\ge\frac{\tau}{2}\Big] &\le \mathbf{Pr}_X\bigg[|\overline{W}-\mathbf{E}_X\big[\overline{W}\big]|\ge\frac{c_0\,s^2\epsilon^2}{n'}\bigg] \le \frac{n'^2\cdot\mathbf{E}_F\Big[8s^2 d_{\max}^{(F)}\Big]}{c_0^2\,s^4\,\epsilon^4} \\
&= \Theta\left(\frac{n'^2\cdot\mathbf{E}_F\Big[d_{\min}^{(F)}\Big]}{s^2\epsilon^4}\right) \le 0.01\,,
\end{aligned}
$$

where the last inequality is true assuming Equation 8 for a sufficiently large constant $c_2$.

**Soundness Case:** In this case $p$ is $\epsilon$-far from $q$. Before showing that $\overline{W}$ is close to zero with high probability, we establish two inequalities as below. First, observe that flattening does not change the $\ell_1$-distance between two distributions due to the following:

$$
\|p-q\|_1 = \sum_{i=i}^n |p(i)-q(i)| = \sum_{i=1}^n\sum_{j=1}^{b_i}\frac{|p(i)-q(i)|}{b_i} = \|p^{(F)}-q^{(F)}\|\,.
$$

Thus, we have the following lower bound for the $\ell_2$-distance between $p^{(F)}$ and $q^{(F)}$ for any $F$:

$$\frac{\epsilon^2}{n'} \leq \frac{\|p-q\|_1^2}{n'} \leq \frac{\|p^{(F)} - q^{(F)}\|_1^2}{n'} \leq \|p^{(F)} - q^{(F)}\|_2^2.$$

Therefore, the above inequality is true in expectation as well:

$$\mathbf{E}_F\left[\|p^{(F)} - q^{(F)}\|_2^2\right] \geq \frac{\epsilon^2}{n'}. \tag{9}$$

Second, we provide the following lemma to show a bound for $\mathbf{E}_F\left[d_{\max}^{(F)}\right]$.

**Lemma A.1.** *Assume $F$ is a random set of samples to be used for flattening. Then, we have:*

$$\mathbf{E}_F\left[d_{\max}^{(F)}\right] \leq \Theta\left(\mathbf{E}_F\left[d_{\min}^{(F)}\right] + \mathbf{E}_F\left[\|p^{(F)} - q^{(F)}\|_2^2\right]\right)$$

For the proof of the lemma, see Section D.

By the Chebyshev inequality and the above lemma, we show that $\overline{W}$ is close to its expectation, i.e., zero, with high probability. Using Jensen's inequality, Equation 5, Equation 9, and Lemma A.1, we have:

$$\mathbf{Pr}_X\left[|\overline{W} - \mathbf{E}_X[\overline{W}]| \geq c_0 \cdot \mathbf{E}_F\left[s^2 \|p^{(F)} - q^{(F)}\|_2^2\right]\right]$$

$$\leq \frac{\mathbf{E}_F\left[8s^3\sqrt{d_{\max}^{(F)}}\|p^{(F)} - q^{(F)}\|_4^2 + 8s^2\, d_{\max}^{(F)}\right]}{c_0^2 \cdot \mathbf{E}_F\left[s^2\|p^{(F)} - q^{(F)}\|_2^2\right]^2}$$

$$\leq \Theta\left(\frac{\sqrt{\mathbf{E}_F\left[d_{\max}^{(F)}\right]} \cdot \sqrt{\mathbf{E}_F\left[\|p^{(F)} - q^{(F)}\|_4^4\right]}}{s \cdot \mathbf{E}_F\left[\|p^{(F)} - q^{(F)}\|_2^2\right]^2} + \frac{\mathbf{E}_F\left[d_{\max}^{(F)}\right]}{s^2 \cdot \mathbf{E}_F\left[\|p^{(F)} - q^{(F)}\|_2^2\right]^2}\right)$$

$$\leq \Theta\left(\frac{\sqrt{\mathbf{E}_F\left[d_{\max}^{(F)}\right]}}{s \cdot \mathbf{E}_F\left[\|p^{(F)} - q^{(F)}\|_2^2\right]} + \frac{\mathbf{E}_F\left[d_{\max}^{(F)}\right]}{s^2 \cdot \mathbf{E}_F\left[\|p^{(F)} - q^{(F)}\|_2^2\right]^2}\right)$$

$$\leq \Theta\left(\frac{\sqrt{\mathbf{E}_F\left[d_{\max}^{(F)}\right]}}{s \cdot \mathbf{E}_F\left[\|p^{(F)} - q^{(F)}\|_2^2\right]}\right)$$

$$\leq \Theta\left(\frac{\sqrt{\mathbf{E}_F\left[d_{\min}^{(F)}\right]} + \sqrt{\mathbf{E}_F\left[\|p^{(F)} - q^{(F)}\|_2^2\right]}}{s \cdot \mathbf{E}_F\left[\|p^{(F)} - q^{(F)}\|_2^2\right]}\right)$$

$$\leq 0.01$$

where the last inequality is true when $s$ is larger than the bound given in Equation 8 below for a sufficiently large constant, $c_2$.

$$s \geq c_2 \cdot \left(\frac{n' \cdot \sqrt{\mathbf{E}_F\left[d_{\min}^{(F)}\right]}}{\epsilon^2}\right) \geq \Theta\left(\frac{n' \cdot \sqrt{\mathbf{E}_F\left[d_{\min}^{(F)}\right]}}{\epsilon^2} + \frac{\sqrt{n'}}{\epsilon}\right) \geq$$

$$\geq \Theta\left(\frac{\sqrt{\mathbf{E}_F\left[d_{\min}^{(F)}\right]}}{\mathbf{E}_F\left[\|p^{(F)} - q^{(F)}\|_2^2\right]} + \frac{1}{\sqrt{\mathbf{E}_F\left[\|p^{(F)} - q^{(F)}\|_2^2\right]}}\right)$$

491 Now, by definition of $\overline{Z}$ and Equation 4, we show $Z$ has to be at least $3\,\tau/2$ with high probability.

492 Therefore, with high probability have:

$$\overline{Z} = \overline{W} + \mathbf{E}_\pi\Big[s^2\|p^{(F)} - q^{(F)}\|_2^2\Big] \geq -c_0 \cdot \mathbf{E}_F\Big[s^2\|p^{(F)} - q^{(F)}\|_2^2\Big] + 4\,c_0 \cdot \mathbf{E}_F\Big[s^2\|p^{(F)} - q^{(F)}\|_2^2\Big]$$

$$\geq 3\,c_0 \cdot \mathbf{E}_F\Big[s^2\|p^{(F)} - q^{(F)}\|_2^2\Big] \geq \frac{3\,c_0\,s^2\epsilon^2}{n'}\,.$$

493 By taking the union bound, the probability of having too large Laplace noise or a too large $|\overline{W}|$ is
494 at most $0.02$. Moreover, Equation 5 and Equation 4 do not hold with probability at most $0.1$. Thus,
495 with probability at least $3/4$, the algorithm output the correct answer. $\qquad\square$

## B   Testing Closeness of Distributions with Unequal Sample Sizes

497 In this section, we prove that the non-private algorithm for testing closeness using unequal sample
498 sizes, provided in [25] with a small modification, is a proper procedure. Therefore, we can turn it into
499 a private tester using our approach provided in Section 3. We also analyze the sample complexity of
500 the tester.

501 Assume $\mathcal{A}$ is the non-private procedure for testing closeness of $p$ and $q$ using unequal sample sizes.
502 First, we explain how $\mathcal{A}$ works. To generate a sample from $p$ (or $q$) the algorithm simply draw an
503 i.i.d. sample from $p$ (or $q$). Assume $k_1$, $k_2$, and $s$ are three parameters that we determine later. $\hat{k}_1$,
504 $\hat{k}_2$, and $\hat{s}$ indicate three Poisson random variables with mean $k_1$, $k_2$ and $s$ respectively. $\mathcal{A}$ draws
505 $\hat{s} + \hat{k}_1$ from $p$ and $\hat{s} + \hat{k}_2$ samples from $q$. For the number of buckets, $\mathcal{A}$ uses the following process.
506 Let $F$ be the number of a set of $\hat{k}_1$ samples from $p$ and $\hat{k}_2$ samples from $q$. The number of buckets
507 for element $i$ is determined by the number of instances of $i$ in $F$ plus one.

508 **Theorem   B.1.**   *There   exists   a   $\xi$-differentially   private   algorithm*
509 *that   uses   $k_1 = \Omega(\max(n^{2/3}/\epsilon^{4/3}, \sqrt{n}/\epsilon\sqrt{\xi}))$   samples   from   $p$,*
510 *$\Theta(\max(n/(\epsilon^2\sqrt{\min(n, k_1)}), \sqrt{n}/\epsilon^2, \sqrt{n}/\epsilon\sqrt{\xi}, 1/\epsilon^2\xi))$ from both $p$ and $q$ and distinguishes*
511 *the following cases with probability at least 0.8:*

512   - *Completeness case: $p = q$*

513   - *Soundness case: $\|p - q\|_1 > \epsilon$.*

514 **Proof:** The goal is to transform the problem to the generate tester we provided in Section 3. First,
515 in Lemma B.2 we show that the non-private algorithm in [25] is a "proper procedure". Using Theo-
516 rem 3.3, the existence of the tester with the sample complexity $s$ for the test part is immediate where
517 $s$ is at least the bound bellow

$$s \geq \Theta\left(\frac{n' \cdot \sqrt{\mathbf{E}_F\big[\min\big(\|p^{(F)}\|_2^2, \|q^{(F)}\|_2^2\big)\big]}}{\epsilon^2} + \frac{\sqrt{n'\Delta(\overline{Z})}}{\epsilon\sqrt{\xi}}\right)\,. \tag{10}$$

We first show the relationship between $s$ above and the rest of the parameters we have. Then we set
the parameters $k_1, k_2$, and $s$ and analyze the sample complexity. Without loss of generality assume
$k_1 \geq k_2$. Note that after flattening the size of the domain increases to $n' = \Theta(n + k_1 + k_2)$ with high
probability. Then, in Lemma B.5, we show that the proposed statistic, $\overline{Z}$, has a bounded sensitivity:

$$\Delta(\overline{Z}) \leq \Theta\left(\frac{k_1}{k_1 + s} \cdot \left(\frac{s + k_2}{k_2}\right)^2\right)\,.$$

In addition, it is shown in [25] that the probability of the expected $\ell_2$-norm of $p$ after flattening is
at most $1/k_1$. Moreover, adding more flattening samples does not increase this quantity. Hence, we
have:

$$\mathbf{E}_F\Big[\min\Big(\|p^{(F)}\|_2^2, \|q^{(F)}\|_2^2\Big)\Big] \leq \frac{1}{k_1}$$

518 We consider the following cases:

- **case 1: $\epsilon = \Omega(\mathbf{n}^{-1/4})$ and $\epsilon^2 \xi = \Omega(\mathbf{n}^{-1})$:** In this case, we have the following properties:

$$\Theta\left(\frac{\sqrt{n}}{\epsilon^2}\right) \leq \Theta\left(\frac{n^{2/3}}{\epsilon^{4/3}}\right) \leq \Theta(n), \quad \text{and} \quad \Theta\left(\frac{1}{\epsilon^2 \xi}\right) \leq \Theta\left(\frac{\sqrt{n}}{\epsilon\sqrt{\xi}}\right) \leq \Theta(n).$$

Let $k_1$ be a number in the range below:

$$\Theta\left(\max\left(\frac{n^{2/3}}{\epsilon^{4/3}} + \frac{\sqrt{n}}{\epsilon\sqrt{\xi}}\right)\right) \leq k_1 \leq \Theta(n).$$

Hence, $n'$ is $\Theta(n)$. Then, we set $s$ and $k_2$ as follows:

$$k_2 := s := \Theta\left(\max\left(\frac{n}{\epsilon^2\sqrt{k_1}}, \frac{\sqrt{n}}{\epsilon\sqrt{\xi}}\right)\right)$$

$\Delta(\overline{Z})$ is $O(1)$ in this case. Therefore, $s$ is $\Omega(n/\epsilon^2\sqrt{k_1} + \sqrt{n\Delta(\overline{Z})}/(\epsilon\sqrt{\xi}))$ and the condition in Equation 10 holds.

- **case 2: $\epsilon = \mathbf{o}(\mathbf{n}^{-1/4})$:** In this case, $\sqrt{n}/\epsilon^2$ is $\Omega(n)$. Thus, we cannot avoid sample complexity of $\Omega(n)$. We set $k_1$ and $k_2$ to be equal to $n$, and we set $s$ to be the following:

$$s := \Theta\left(\max\left(\frac{\sqrt{n}}{\epsilon^2}, \frac{1}{\epsilon^2 \xi}\right)\right).$$

Clearly $n'$ is still $\Theta(n)$, and $s$ is $\Omega(n/\sqrt{k_1}\epsilon^2)$. In this case $\Delta(\overline{Z})$ is $\Theta(s/n)$. Hence, in order to have $s$ at least $\Omega\sqrt{n'\Delta(\overline{Z})}/\epsilon\sqrt{\xi}$, it suffices to have $s = \Omega(1/\epsilon\sqrt{\xi})$.

$\square$

## B.1 Non-Private Closeness Tester Is a Proper Procedure

**Lemma B.2.** *Procedure $\mathcal{A}$ explained above is a proper procedure according to Definition 3.2.*

**Proof:** First, we show the number of samples we generate is not too far from their expectation. Hence $X$ can be a set with a bounded number of samples. In the following lemma we show if the means are larger than a fixed constant, then with probability $0.01$ we can assume the number of samples from each of distributions is at most three times larger than their means.

**Lemma B.3.** *Assume random variable $x$ is drawn from $\mathbf{Poi}(\lambda)$. If $\lambda$ is at least $1.5 \cdot \ln(1/c)$, then the probability of $x$ being larger than $3\lambda$ is at most $1 - c$.*

Now, we only need to show that inequalities in Equation 4 and Equation 5 are correct. Before proving the equations, we provide an insightful information about the distribution over the $b_i$'s. It is clear that for a fixed $i$, $b_i - 1$ is an independent Poisson random variable with mean $k_1 p(i) + k_2 q(i)$. More precisely, we can think of $b_i - 1$ as the sum of two random variables $b_{i,1} \sim \mathbf{Poi}(k_1 p(i))$ and $b_{i,2} \sim \mathbf{Poi}(k_2 q(i))$ plus one. However, assume a random set of samples, $X$, is given to us with $t_{i,1}$ instances of $i$ from $p$, and $t_{i,2}$ instances of $i$ from $q$. Then, considering a random permutation of samples, then $b_{i,j}$ is a binomial random variable from $\mathbf{Bin}(t_{i,j}, k_j/(k_j + s))$ for $j = 1, 2$.

Now, we focus on proving Equation 4. Fix a set of sample $X$ and a domain element $i$. Using Jensen's inequality, we have

$$\mathbf{E}_\pi\left[\frac{1}{b_i(X,\pi)}\right] = \mathbf{E}_{b_{i,1},b_{i,2}}\left[\frac{1}{b_i}\right] \geq \frac{1}{\mathbf{E}_{b_{i,1},b_{i,2}}[b_i]} = \frac{1}{t_{i,1}\, k_1/(k_1 + s) + t_{i,2}\, k_2/(k_2 + s) + 1}.$$

Note that by Markov's inequality the probability of any of $t_{i,1}$ or $t_{i,2}$ being 50 times[3] larger than their expectations is at most $0.04$. Therefore, with probability $0.96$ assume they are at most 50 times their expectation. Since $t_{i,1}$ and $t_{i,2}$ are two Poisson random variables with means $p(i)(k_1 + s)$ and $q(i)(k_2 + s)$, we can bound the above quantity as follows:

$$\mathbf{E}_\pi\left[\frac{1}{b_i(\pi)}\right] \geq \frac{1}{50\, p(i)k_1 + 50\, q(i)k_2 + 1} = \frac{1}{50\lambda + 1}$$

where we use $\lambda$ to denote $p(i)k_1 + q(i)k_2$. On the other hand, the expected value of $1/b_i$ when $X$ has not be observed is the following:

$$\mathbf{E}_F\left[\frac{1}{b_i}\right] = \mathbf{E}_{x\sim\mathbf{Poi}(\lambda)}\left[\frac{1}{x+1}\right] = \frac{1-e^{-\lambda}}{\lambda} \leq \frac{65}{50\lambda+1}\,.$$

545 Putting all of the above facts together, we conclude:

$$\mathbf{Pr}_{t_{i,1},t_{i,2}}\left[\mathbf{E}_\pi\left[\frac{(p(i)-q(i))^2}{b_i(\pi)}\right] \geq \frac{1}{65}\cdot\mathbf{E}_F\left[\frac{(p(i)-q(i))^2}{b_i}\right]\right] \geq 0.96\,. \tag{11}$$

We define a random variable $x_i$ over the randomness of $t_{i,1}$ and $t_{i,2}$ to be the following:

$$x_i := \mathbf{E}_\pi\left[\frac{(p(i)-q(i))^2}{b_i(\pi)}\right]$$

546 Note that by the *Poissonization method*, all the number of instances of a particular element are
547 independent form the rest. Hence, $x_i$'s are independent given the independence of $t_{i,j}$'s. In addition,
548 we prove the following lemma, to bound the sum of $x_i$'s from below:

549 **Lemma B.4.** *Assume we have $n$ independent random variables $x_1, x_2, \ldots, x_n$ in the range $[0, +\infty)$.*
550 *Suppose each $x_i$ is at least $A_i$ with probability $p \geq 0.95$ where $A_i$ is a fixed number. Then, with*
551 *probability at least 0.9, $\sum_{i=1}^n x_i$ is at least $0.1\sum_{i=1}^n A_i$.*

552 For the proof of the lemma, see Section D.

553 Using Equation 11, and Lemma B.4, with probability 0.9 we have:

$$\mathbf{E}_\pi\left[\|p^{(F)}-q^{(F)}\|_2^2\right] = \mathbf{E}_\pi\left[\frac{(p(i)-q(i))^2}{b_i(\pi)}\right] \geq \frac{1}{650}\cdot\sum_{i=1}^n\mathbf{E}_F\left[\frac{(p(i)-q(i))^2}{b_i}\right]$$

$$= 4c_0\cdot\mathbf{E}_F\left[\|p^{(F)}-q^{(F)}\|_2^2\right]^2\,.$$

554 where $c_0 = 1/26000$. Hence, the proof of Equation 4 is complete.

555 Now, we focus on proving Equation 5. To prove the inequality, it suffices to show that $\mathbf{E}_F\left[1/b_i^3\right]$ is
556 $O\left(\mathbf{E}_F[1/b_i]^2\right)$. Note one can think of $b_i$ to be equal to $x+1$ where $x$ is a Poisson random variable
557 with mean $\lambda' = p(i)k_1 + q(i)k_2$. Thus, we have:

$$\mathbf{E}_F\left[\frac{1}{b_i^3}\right] = \mathbf{E}_{x\sim\mathbf{Poi}(\lambda')}\left[\frac{1}{(x+1)^3}\right] \leq \mathbf{E}\left[\frac{6}{(x+1)(x+2)(x+3)}\right] \leq 6\cdot\sum_{x=0}^\infty\frac{e^{-\lambda'}\lambda'^x}{(x+3)!}$$

$$= \frac{6}{\lambda'^3}\sum_{y=3}^\infty\frac{e^{-\lambda'}\lambda'^y}{y!} = \frac{6\left(1-e^{-\lambda'}-e^{-\lambda'}\lambda'-e^{-\lambda'}\lambda'^2/2\right)}{\lambda'^3} \leq 6\cdot\left(\frac{1-e^{-\lambda'}}{\lambda'}\right)^2\,. \tag{12}$$

558 On the other hand, we can compute the expected value of $1/b_i$ as follows:

$$\left(\mathbf{E}_F\left[\frac{1}{b_i}\right]\right)^2 = \left(\sum_{x=0}^\infty\frac{e^{-\lambda'}\lambda'^x}{(x+1)!}\right)^2 = \left(\frac{1}{\lambda'}\sum_{y=1}^\infty\frac{e^{-\lambda'}\lambda'^y}{y!}\right)^2 = \left(\frac{1-e^{-\lambda'}}{\lambda'}\right)^2\,.$$

559 Putting these two equations together, one can conclude the Equation 5:

$$\mathbf{E}_F\left[\|p^{(F)}-q^{(F)}\|_4^4\right] = \sum_{i=1}^n\mathbf{E}_F\left[\frac{(p(i)-q(i))^4}{b_i^3}\right] \leq 6\cdot\sum_{i=1}^n\left(\mathbf{E}_F\left[\frac{(p(i)-q(i))^2}{b_i}\right]\right)^2$$

$$\leq 6\cdot\left(\sum_{i=1}^n\mathbf{E}_F\left[\frac{(p(i)-q(i))^2}{b_i}\right]\right)^2 = 6\cdot\mathbf{E}_F\left[\|p^{(F)}-q^{(F)}\|_2^2\right]^2\,.$$

560 Therefore, the statement of the lemma is concluded. $\qquad\square$

 **B.2 Bounding the Sensitivity**

In this section, we provide an upper bound for the amount that the statistic changes if we change one sample in the input. In other words, we find an upper bound for the sensitivity of $\overline{Z}$ as define in Equation 2.

We start off with defining the notation we use in this section. Let $X = (X_1, X_2)$ be a set of samples consist of $m_1$ samples from $p$ and $m_2$ samples from $q$. As we had before, $\hat{k}_j = \mathbf{Poi}(k_j)$ and $\hat{s} = \mathbf{Poi}(s)$ to denote the number of samples for test and the flattening respectively. Note that now that $m_1 = \hat{k}_j + \hat{s}$ is observed, then $\hat{k}_j$ is a binomial random variable, $\mathbf{Bin}(m, k_j/(k_j + s))$. Let $F_1$ and $F_2$ denote the set of sample from distribution $p$ and $q$ we use for the flattening. Given $X$, $F_1$ and $F_2$ are determined by the $\hat{k}_j$'s, and the order of the samples which is determined by $\pi$. Although the $F_j$'s are two sets of ordered samples, the order of the element in them does not matter. We may consider them equivalent to set of *indices* $I_j$, such that the $r$-th sample is in $F_j$ if and only if $r$ is in $I_j$. In this sense, we can define the probability of an index set $I_j$ to be the probability of taking a flattening set $F_j$, such that $F_j$ is equivalent to $I_j$. The randomness in this probability is taken over the choice of the permutation $\pi$ and $\hat{k}_j$. In addition, for each element $i$, we use the following notation:

- $t_{i,j}$ is the number of instances of element $i$ in the set $X_j$.

- $k_{i,j}$ is the number of instances of element $i$ in the set $F_j$.

- $s_{i,j}$ is the number of instances of element $i$ in the set $X_j \setminus F_j$.

Based on these notations, the statistic $\overline{Z}$ is the following, we use the equivalent intermediate statistic as indicated in Lemma **??**, and denote it by $z_i(X, I_1, I_2)$

$$\overline{Z} = \mathbf{E}_{\pi,r}[Z] = \sum_{I_1, I_2} \mathbf{Pr}[I_1] \cdot \mathbf{Pr}[I_2] \cdot \sum_{i=1}^{n} z_i(X, I_1, I_2)$$

$$= \sum_{I_1, I_2} \mathbf{Pr}[I_1] \cdot \mathbf{Pr}[I_2] \cdot \sum_{i=1}^{n} \frac{(s_{i,1} - s_{i,2})^2 - s_{i,1} - s_{i,2}}{k_{i,1} + k_{i,2} + 1}$$

Assume $X' = (X_1', X_2')$ is also a set of samples such that it differs in exactly one sample compared to $X$. We similarly define all the notation for $X'$ as well by adding the prime notation to each letter. Without loss of generality, we assume that the $r$-th sample in $X_1$, namely $x_r$, is different from the $r$-th sample in $X_1'$, namely $x_r'$, and all other samples are the same. Now, we are ready to bound the sensitivity of $\overline{Z}$ in the following lemma:

**Lemma B.5.** *For two neighboring sample sets, $X_1$, $X_2$, and $X_1'$, $X_2'$, and for fixed $k_1$ and $k_2$, we have:*

$$|\overline{Z} - \overline{z}'| \le \Theta\left(\frac{k_1}{k_1 + s} \cdot \left(\frac{s + k_2}{k_2}\right)^2 + \frac{k_2}{k_2 + s} \cdot \left(\frac{s + k_1}{k_1}\right)^2\right)$$

**Proof:** By the definition of $\overline{Z}$, by the triangle inequality and Bayes' law, we can find an upper bound for the difference of the statistics as follows:

$$|\overline{Z} - \overline{z}'| = \left| \sum_{I_1, I_2} \mathbf{Pr}[I_1] \cdot \mathbf{Pr}[I_2] \cdot \sum_{i=1}^{n} z_i(X, I_1, I_2) - z_i'(X', I_1, I_2) \right|$$

$$\leq \sum_{i=1}^{n} \sum_{I_2} \mathbf{Pr}[I_2] \cdot \sum_{I_1} \mathbf{Pr}[I_1] \cdot |z_i(X_1, X_2, I_1, I_2) - z_i'(X_1', X_2', I_1, I_2)|$$

$$\leq \sum_{i=1}^{n} \sum_{I_2} \mathbf{Pr}[I_2] \cdot \sum_{I_1} \mathbf{Pr}[I_1 | r \in I_1] \cdot \mathbf{Pr}[r \in I_1] \cdot |z_i(X, I_1, I_2) - z_i'(X', I_1, I_2)|$$

$$+ \sum_{i=1}^{n} \sum_{I_2} \mathbf{Pr}[I_2] \cdot \sum_{I_1} \mathbf{Pr}[I_1 | r \notin I_1] \cdot \mathbf{Pr}[r \notin I_1] \cdot |z_i(X, I_1, I_2) - z_i'(X', I_1, I_2)| \ .$$

$$(13)$$

It is clear that if $i$ is a domain element which none of its instances in the sample set is changed, then the number of occurrences of $i$ remains unchanged in the same subsets of the $X_j$'s and the $X_j'$'s. Thus, $z_i - z_i'$ is equal to zero for $i \notin \{x_r, x_r'\}$. For now, we assume $i$ is equal to $x_r$. One can replicate the same bound when $i = x_r'$.

It is clear that $t_{i,1} = t_{i,1}' + 1$. Fix a subset of indices, $I_2 \subseteq [m_2]$. Since $X_2$ and $X_2'$ are the same, then $s_{i,2} = s_{i,2}'$ and $k_{i,2} = k_{i,2}'$. Let $\ell$ be an index in $\{1, 2\}$ such that $t_{i,\ell}$ denotes the maximum of $t_{i,1}$ and $t_{i,2}$. Observe that we always have $s_{i,j} = t_{i,j} - k_{i,j}$ by definition. Now, we rewrite the difference of the $z_i$ and $z_i'$ as below using the triangle inequality:

$$|z_i(X, I_1, I_2) - z_i'(X', I_1, I_2)| = \left| \frac{(s_{i,1} - s_{i,2})^2 - s_{i,1} - s_{i,2}}{k_{i,1} + k_{i,2} + 1} - \frac{(s_{i,1}' - s_{i,2}')^2 - s_{i,1}' - s_{i,2}'}{k_{i,1}' + k_{i,2}' + 1} \right|$$

$$= \left| \frac{(s_{i,1} - s_{i,2})^2 - s_{i,1} - s_{i,2}}{k_{i,1} + k_{i,2} + 1} - \frac{(s_{i,1}' - s_{i,2})^2 - s_{i,1}' - s_{i,2}}{k_{i,1}' + k_{i,2} + 1} \right|$$

$$\leq \left| \frac{(s_{i,1} - s_{i,2})^2 - s_{i,1} - s_{i,2}}{k_{i,1} + k_{i,2} + 1} - \frac{(s_{i,1}' - s_{i,2})^2 - s_{i,1}' - s_{i,2}}{k_{i,1} + k_{i,2} + 1} \right|$$

$$+ \left| \frac{(s_{i,1}' - s_{i,2})^2 - s_{i,1}' - s_{i,2}}{k_{i,1} + k_{i,2} + 1} - \frac{(s_{i,1}' - s_{i,2})^2 - s_{i,1}' - s_{i,2}}{k_{i,1}' + k_{i,2} + 1} \right|$$

Note that if $r$ is in $I_1$, the number of instance of $x_r$ in the flattening set changes. More precisely, $k_{i,1}$ is $k_{i,1}' + 1$. However, $s_{i,1}$ remains equal to $s_{i,1}'$ and the changed sample does not affect it, so the second to the last line above is zero. Similarly, if $r$ is not in $I_1$, then $k_{i,1}$ remains the same as $k_{i,1}'$, and $s_{i,1} = s_{i,1}' + 1$, so the last line above will be zero. $s_{i,j}$ and $k_{i,j}$ are at most $t_{i,j}$ by definition Therefore, if we use the fact that $s_{i,j}$ and $k_{i,j}$ are at most $t_{i,j}$ by definition, then we have:

$$\sum_{I_1} \mathbf{Pr}[I_1] \cdot |z_i(X, I_1, I_2) - z_i'(X', I_1, I_2)|$$

$$\leq \sum_{I_1} \mathbf{Pr}[I_1 | r \in I_1] \cdot \mathbf{Pr}[r \in I_1] \cdot 2\, t_{i,\ell}^2 \cdot \left| \frac{1}{k_{i,1} + k_{i,2} + 1} - \frac{1}{k_{i,1}' + k_{i,2} + 1} \right|$$

$$+ \sum_{I_1} \mathbf{Pr}[I_1 | r \notin I_1] \cdot \mathbf{Pr}[r \notin I_1] \cdot \frac{2\, t_{i,\ell} \cdot}{k_{i,1} + k_{i,2} + 1} \ .$$

Using the properties of the Poissonization method, given that we observed $t_{i,j} = k_{i,j} + s_{i,j}$, then $k_{i,j}$ is Binomial random variable: $\mathbf{Bin}(t_{i,j}, k_j/(k_j + s))$. Given this fact, the probability of $r \in I$

is $k_1/(k_1 + s)$. Therefore, we have:

$$\sum_{I_2} \mathbf{Pr}[I_2] \cdot \sum_{I_1} \mathbf{Pr}[I_1] \cdot |z_i(X, I_1, I_2) - z_i'(X', I_1, I_2)|$$

$$= \sum_{I_2} \mathbf{Pr}[I_2] \cdot \sum_{I_1} \mathbf{Pr}[I_1|r \in I_1] \cdot \frac{k_1}{k_1 + s} \cdot \frac{2\, t_{i,\ell}^2}{(k_{i,1}' + k_{i,2} + 2) \cdot (k_{i,1}' + k_{i,2} + 1)}$$

$$+ \sum_{I_2} \mathbf{Pr}[I_2] \cdot \sum_{I_1} \mathbf{Pr}[I_1|r \notin I_1] \cdot \frac{s}{k_1 + s} \cdot \frac{2\, t_{i,\ell}}{k_{i,1} + k_{i,2} + 1}$$

$$\leq \frac{2\, k_1\, t_{i,\ell}^2}{k_1 + s} \cdot \sum_{I_2} \mathbf{Pr}[I_2] \cdot \sum_{I_1} \mathbf{Pr}[I_1|r \in I_1] \cdot \frac{1}{(k_{i,\ell}' + 2) \cdot (k_{i,\ell}' + 1)}$$

$$+ \frac{2\, s\, t_{i,\ell}}{k_1 + s} \cdot \sum_{I_2} \mathbf{Pr}[I_2] \cdot \sum_{I_1} \mathbf{Pr}[I_1|r \notin I_1] \cdot \frac{1}{k_{i,\ell} + 1}$$

$$\leq \frac{2\, k_1\, t_{i,\ell}^2}{k_1 + s} \cdot \mathbf{E}_{k_{i,\ell}' \sim \mathbf{Bin}(t_{i,j}-1, k_1/(k_1+s))} \left[ \frac{1}{(k_{i,\ell}' + 2)(k_{i,j}' + 1)} \right]$$

$$+ \frac{2\, s\, t_{i,\ell}}{k_1 + s} \cdot \mathbf{E}_{k_{i,\ell} \sim \mathbf{Bin}(t_{i,j}-1, k_1/(k_1+s))} \left[ \frac{1}{k_{i,\ell} + 1} \right].$$

Using Lemma D.2 and Lemma D.3, we have:

$$\sum_{I_2} \mathbf{Pr}[I_2] \cdot \sum_{I_1} \mathbf{Pr}[I_1] \cdot |z_i(X, I_1, I_2) - z_i'(X', I_1, I_2)|$$

$$\leq \frac{2\, k_1}{k_1 + s} \cdot \left( \frac{s + k_\ell}{k_\ell} \right)^2 + \frac{2\, s}{k_1 + s} \cdot \frac{s + k_\ell}{k_\ell}.$$

Note that $\ell$ can be one or two. Also, the upper bound for $\overline{Z} - \overline{z}'$ is twice as above, since we have to consider the case when $i = x_r'$. Moreover, the bound we get is based on this assumption that the changed sample is in $X_1$, so to find the upper bound we need to consider both cases. Hence, we have the following bound:

$$|\overline{Z} - \overline{z}'| \leq \Theta \left( \frac{k_1}{k_1 + s} \cdot \left( \frac{s + k_2}{k_2} \right)^2 + \frac{k_2}{k_2 + s} \cdot \left( \frac{s + k_1}{k_1} \right)^2 \right)$$

and the proof is complete. $\qquad\square$

# C   Testing independence of two random variables

In this section, we provide a $\xi$-differentially private tester for testing independence of two random variables. The idea is to reduce the optimal non-private tester, delivered in [25], to a private one using the technique we explained in Section 3.

We start off with defining the problem and the non-private procedure $\mathcal{A}$ that reduces testing independence to the testing closeness of two distributions. Assume $p$ is a distribution over $[n] \times [m]$. Without loss of generality, we assume $m \leq n$. Suppose we receive samples $(x, y)$ from $p$. We say distribution $p$ is an independent distribution, if the $x$'s and the $y$'s are independent from each other. The goal is to distinguish whether $p$ is an independent distribution or is it $\epsilon$-far from any independent distribution over $[n] \times [m]$. It is known that if $p$ is an independent distribution, then $p$ is equal to $p_1 \times p_2$, and if $p$ is $epsilon$-far from being independent, then $p$ is $\epsilon/3$-far from $p_1 \times p_2$ where $p_1$ and $p_2$ are the marginal distributions of $p$. Using this fact, the non-private tester reduces the problem to testing the closeness of $p$ and $q := p_1 \times p_2$ [6].

Here, we describe a proper procedure, say $\mathcal{A}$, for reducing testing independence of $p$ to the testing closeness of $p$ and $q$, so it can be turned into a private algorithm using the method explained in Section 3. First, we describe the sampling scheme of the procedure: For every sample that the algorithm needs, it draws two samples, and puts them in a *block*. We denote a block of two samples

$(x_1, y_1)$ and $(x_2, y_2)$ by $\langle (x_1, y_1), (x_2, y_2) \rangle$. We can use the samples in block to obtain a sample from $p$, $p_1$, $p_2$, and $q$ as follows. To get a sample from $p$, we always take the first samples, $(x_1, y_1)$, in the block. We take $x_1$, $y_2$ as two the samples from $p_1$ and $p_2$. In addition, since $x_1$ and $y_2$ are two independent random variables, $(x_1, y_2)$ is a sample from $q$. Also, we use "dot notation" to indicate an arbitrary element in the domain. For example, for a given $x$, $(x, .)$ is a sample that its first coordinate is $x$ and the second coordinate can be any $y$ in $[m]$. Similarly, we use the same notation to refer to a block, for example for a given $x$ and $y$, $\langle (x, .), (., y) \rangle$ indicates a block that the first coordinate of the first sample is $x$ and the second coordinate of the second sample is $y$, and the two other coordinates can be arbitrary elements in $[m]$ and $[n]$. Let $X$ denotes the set of all blocks that are available to the procedure. We use $f$ to denote a frequency of the blocks with a certain format in $X$., e.g., $f_{\langle (x, .), (., .) \rangle}$ is the number of blocks in $X$ that the first coordinate of the first sample is $x$. For the rest of this section, we focus on blocks and use them accordingly to extract a sample.

Suppose we have sample access to $p$, and we can draw blocks of samples from it. Procedure $\mathcal{A}$ uses the blocks for the following purposes: the *flattening samples* are used to determine the number of buckets for each domain element. They are designed to make sure that the $\ell_2$-norm of $q$ after flattening is low. Also, the *test samples* are used to generate samples from two distributions $p$ and $q$, which we test their closeness. Below is how the algorithm will determine these samples. For now, assume $k^{(p_1)}$, $k^{(p_2)}$, $k^{(p)}$, $k^{(q)}$, and $s$ are parameters that we determine later.

**Flattening samples and the number of buckets:**  We flatten distribution $p$ using samples from the marginal distributions $p_1$ and $p_2$, and also samples from $p$ itself. More specifically, we draw four sets of blocks from $p$, namely $F^{(p_1)}, F^{(p_2)}, F^{(p)}, F^{(q)}$, which contain $\mathbf{Poi}\left(k^{(p_1)}\right)$, $\mathbf{Poi}\left(k^{(p_2)}\right)$, $\mathbf{Poi}\left(k^{(p)}\right)$, $\mathbf{Poi}\left(k^{(q)}\right)$ blocks respectively. We refer to the samples in these sets as flattening samples, and denote the collection of them by $F$. As we discuss earlier, we extract samples from the blocks in these sets to obtain samples from $p_1, p_2, p$, and $q$. More specifically, we use the following notation for the number of occurrences of each sample obtained from each set:

- $k_x^{(p_1)}$ denotes the number of occurrences of the blocks of the form $\langle (x, .), (., .) \rangle$ in the flattening set $F^{(p_1)}$.

- $k_y^{(p_2)}$ denotes the number of occurrences of the blocks of the form $\langle (., .), (., y) \rangle$ in the flattening set $F^{(p_2)}$.

- $k_{(x,y)}^{(p)}$ denotes the number of occurrences of the blocks of the form $\langle (x, y), (., .) \rangle$ in the flattening set $F^{(p)}$.

- $k_{(x,y)}^{(q)}$ denotes the number of occurrences of the blocks of the form $\langle (x, .), (., y) \rangle$ in the flattening set $F^{(q)}$.

Our procedure uses $b_{(x,y)}$ many buckets for a domain element $(x, y)$, where $b_{(x,y)}$ is defined as follows:
$$b_{(x,y)} = (k_x^{(p_1)} + 1)(k_y^{(p_2)} + 1) + k_{(x,y)}^{(p)} + k_{(x,y)}^{(q)}.$$

It is worth to note that $k_x^{(p_1)}$ is always determined by the first samples in the blocks, whereas $k_y^{(p_2)}$ is determined by the second samples in the blocks. Therefore, for all $x$'s and $y$'s, these quantities are independent of each other.

**Test samples:**  To determine the test samples, we draw two sets of blocks $T^{(p)}$ and $T^{(q)}$. Each set contains $\mathbf{Poi}(s)$ many blocks. The samples in $T^{(p)}$ and $T^{(q)}$ are our test samples, and we denote the collection of these two sets by $T$. The blocks in $T^{(p)}$ are used to obtain samples form $p$, and the blocks in $T^{(q)}$ are used to collect samples from $q$. In particular, we use the following notation for the number of occurrences of each domain element $(x, y)$.

- $s_{(x,y)}^{(p)}$ denotes the number of occurrences of the blocks of the form $\langle (x, y), (., .) \rangle$ in the test set $T^{(p)}$.

- $s_{(x,y)}^{(q)}$ denotes the number of occurrences of the blocks of the form $\langle (x, .), (., y) \rangle$ in the test set $T^{(q)}$.

Now, that we showed how procedure $\mathcal{A}$ determines the number of samples, we prove it yields to a $\xi$-differentially private tester as well. At a high level, we first show that $\mathcal{A}$ is a proper procedure for testing independence (Section C.1), then use our general closeness tester to achieve a $\xi$-differentially private tester. Since the sample complexity of the private tester depends on the sensitivity of the statistic we are using, we analyze the sensitivity of the statistic (Section C.2). In particular, we show if the number of occurrences of certain blocks in the sample set is "as expected," then the sensitivity is low, which results in a nearly optimal sample complexity for the private tester. Next, we develop a framework to extend the input domain of the private algorithm to any sample set (Section C.3). More formally, we have the following theorem:

**Theorem C.1.** *Let $p$ be a distribution over $[n] \times [m]$. There exists a $\xi$-differentially private $(\epsilon, 2/3)$ tester for the testing independence of $p$ that uses $\Theta(s)$ samples where $s$ is:*

$$s = \Theta \left( \frac{n^{2/3} m^{1/3}}{\epsilon^{4/3}} + \frac{(m\,n)^{1/2}}{\epsilon^2} + \frac{(m\,n \log n)^{1/2}}{\epsilon \sqrt{\xi}} + \frac{\log n}{\epsilon^2 \xi} \right).$$

**Proof:** We first set up the parameters we use:

$$k^{(p_2)} = m, \quad k^{(p_1)} = \min(n, n^{2/3} m^{1/3}/\epsilon^{4/3}), \quad k^{(p)} = k^{(q)} = \min(m \cdot n, s),$$

$$\text{and} \quad s = c \cdot \left( \frac{n^{2/3} m^{1/3}}{\epsilon^{4/3}} + \frac{(m\,n)^{1/2}}{\epsilon^2} + \frac{(m\,n \log n)^{1/2}}{\epsilon \sqrt{\xi}} + \frac{\log n}{\epsilon^2 \xi} \right).$$

where $c$ is a large constant. For sufficiently large $m$ and $n$, with probability 0.99 the number of blocks in each set in $\mathcal{S} = \{F^{(p_1)}, F^{(p_2)}, F^{(p)}, F^{(q)}, T^{(p)}, T^{(q)}\}$ is within a constant factor of its expectation via Lemma B.3. Now, we show that $\mathcal{A}$ that we describe above is a proper procedure:

**Lemma C.2.** *Procedure $\mathcal{A}$ explained above is a proper procedure according to Definition 3.2 for testing independence of two random variable.*

The proof of the above Lemma is in Section C.1. Now, using Theorem 3.3, there exists a $\xi$-differentially private tester for the independence property which uses the following number of test samples:

$$s' := \Theta \left( \frac{n' \cdot \sqrt{\mathbf{E}_F \left[ \min \left( \|p^{(F)}\|_2^2, \|q^{(F)}\|_2^2 \right) \right]}}{\epsilon^2} + \frac{\sqrt{n' \Delta(\overline{Z})}}{\epsilon \sqrt{\xi}} \right).$$

Here, we show that $s \geq s'$, thus the number of samples that the procedure provides is enough by bounding $n'$, $\mathbf{E}_F \left[ \min \left( \|p^{(F)}\|_2^2, \|q^{(F)}\|_2^2 \right) \right]$, and $\Delta(\overline{Z})$. Note that $n'$ is the new domain size which is equal to $\sum_{(x,y)} b_{(x,y)} = \Theta(m\,n)$. The expected of minimum of the $\ell_2$-norm of $p$ and $q$ is bounded using the result in Lemma 2.6 in [25].

$$\mathbf{E}_{F^{(p_1)}, F^{(p_2)}, F^{(p)}} \left[ \min \left( \|p^{(F)}\|_2^2, \|q^{(F)}\|_2^2 \right) \right] \leq \mathbf{E}_F \left[ \|q^{(F)}\|_2^2 \right] \leq \sum_{x=1}^{n} \sum_{y=1}^{m} \frac{q(x,y)^2}{b_{(x,y)}}$$

$$\leq \sum_{x=1}^{n} \sum_{y=1}^{m} \frac{p_1(x)^2 \, p_2(y)^2}{(k_x^{(p_1)} + 1) \cdot (k_y^{(p_2)} + 1)} \leq \|p_1^{(F^{(p_1)})}\| \cdot \|p_2^{(F^{(p_2)})}\| \leq \frac{1}{k^{(p_1)} \, k^{(p_2)}}.$$

Moreover, in Section C.2, we provide the following bound for the sensitivity of the statistic:

**Lemma C.3.** *Given that the size of all flattening and test samples are within the constant factor of their expectations, the sensitivity of the statistic $Z$ is bounded as follows:*

$$\Theta \left( \frac{s}{k^{(q)}} + \frac{s}{k^{(p)}} + \frac{s}{k^{(p)}} \cdot \frac{f_{\langle (.,b),(.,.) \rangle}}{f_{\langle (.,.),(.,b) \rangle} + 1} \right)$$

To get a bound on sensitivity, for now, suppose all the input block sets $X$ has a desired property that the ratio between $f_{\langle (.,b),(.,.) \rangle}$ and $f_{\langle (.,.),(.,b) \rangle} + 1$ are bounded:

$$X \in \mathcal{X}^* := \left\{ X : \frac{f_{\langle (.,b),(.,.)\rangle}}{f_{\langle (.,.),(.,b)\rangle} + 1} \leq \tau \right\}$$

where $\tau = 1200 \ln n$. Thus, using the fact that $X$ is in $\mathcal{X}^*$, one can obtain:

$$\Delta(\overline{Z}) \leq \Theta\left(\frac{s \log n}{mn} + \log n\right)$$

Now, we are ready to show that $s' \leq s$ implying that we have enough samples for the $\xi$-private tester. It is not hard to see that we have the following bounds (up to a constant factors):

$$
\begin{aligned}
s' &\leq \Theta\left(\frac{n' \cdot \sqrt{\mathbf{E}_F\left[\min\left(\|p^{(F)}\|_2^2, \|q^{(F)}\|_2^2\right)\right]}}{\epsilon^2} + \frac{\sqrt{n'\Delta(\overline{Z})}}{\epsilon\sqrt{\xi}}\right) \\
&\leq \Theta\left(\frac{m\,n}{\epsilon^2\sqrt{k^{(p_1)}\,k^{(p_2)}}} + \frac{\sqrt{m\,n\Delta(\overline{Z})}}{\epsilon\sqrt{\xi}}\right) \\
&\leq \Theta\left(\frac{m\,n}{\epsilon^2\sqrt{m\,\min(n, n^{2/3}m^{1/3}/\epsilon^{4/3})}} + \frac{\sqrt{m\,n}}{\epsilon\sqrt{\xi}} \cdot \sqrt{\frac{s\log n}{mn} + \log n}\right) \\
&\leq \Theta\left(\frac{n^{2/3}m^{1/3}}{\epsilon^{4/3}} + \frac{(m\,n)^{1/2}}{\epsilon^2} + \frac{\sqrt{m\,n}}{\epsilon\sqrt{\xi}} \cdot \sqrt{\frac{s\log n}{mn} + \log n}\right) \\
&\leq s
\end{aligned}
$$

Thus, given that $X$ is in $\mathcal{X}^*$, there exists a $\xi$-differentially private tester that outputs the right answer with probability 0.8. This is sufficient to show that there exists an $\xi$-differentially private algorithm with asymptotically the same number of samples via Lemma C.6. $\qquad\square$

## C.1 Non-Private Independence Tester is a proper procedure

**Lemma C.2.** *Procedure $\mathcal{A}$ explained above is a proper procedure according to Definition 3.2 for testing independence of two random variable.*

**Proof:** Let $X$ be the set of all blocks we received. Since the number of blocks in each of the flattening set and the test set is Poisson random variable, by Lemma B.3, we can conclude with probability 1-0.01 we draw at most three times more samples than what is expected. Hence $X$ is a set with a bounded number of samples.

We start proving Equation 4 by recalling a fact about the Poissonization method. The number of blocks of a certain form in one of the flattening and test sets is a Binomial random variable with the bias that is proportional to the expected size of each set. For example, if $X$ contains $t$ blocks of the form $\langle (x,.), (.,y)\rangle$, the number of the blocks of the form $\langle (x,.), (.,y)\rangle$ in $T^{(q)}$, namely $r$, is $\mathbf{Bin}(t, s/(k^{(p_1)} + k^{(p_2)} + k^{(p)} + 2s))$. Moreover, the probability of getting a block of this type is $q(x,y) = p_1(x) \cdot p_2(y)$, so $t$ is a Poisson random variable with mean $q(x,y) \cdot (k^{(p_1)} + k^{(p_2)} + k^{(p)} + 2s)$ over the randomness of $X$. By Markov's inequality, with probability $1 - 1/c$, we may assume $t$ is at most $c$ times its expectation. As a consequence, $\mathbf{E}[r]$ is at most $c \cdot \mathbf{E}[t] \cdot s/(k^{(p_1)} + k^{(p_2)} + k^{(p)} + 2s) = c\,q(x,y)\,s$. Note that we can extend this example further to any type of block and any test or flattening sets.

Given $X$, for a domain element $(x, y)$, the following holds using Jensen's inequality:

$$\mathbf{E}_\pi\left[\frac{1}{b_{(x,y)}(X,\pi)}\right] = \mathbf{E}_{k_x^{(p_1)},k_y^{(p_2)},k_{(x,y)}^{(p)}}\left[\frac{1}{b_{(x,y)}}\right] \geq \frac{1}{\mathbf{E}_{k_x^{(p_1)},k_y^{(p_2)},k_{(x,y)}^{(p)}}\left[b_{(x,y)}\right]}$$

$$= \frac{1}{(\mathbf{E}\left[k_x^{(p_1)}\right]+1)\cdot(\mathbf{E}\left[k_y^{(p_2)}\right]+1)+\mathbf{E}\left[k_{(x,y)}^{(p)}\right]}$$

$$\geq \frac{1}{(50\,p_1(x)\,k^{(p_1)}+1)\cdot(50\,p_2(y)\,k^{(p_2)}+1)+100\,p(x,y)\,k^{(p)}}$$

$$\geq \frac{1}{2500}\cdot\frac{1}{(p_1(x)\,k^{(p_1)}+1)\cdot(p_2(y)\,k^{(p_2)}+1)+p(x,y)\,k^{(p)}}\,.$$

where the second to last inequality holds with probability 0.95.

On the other hand, we find an upper bound of $\mathbf{E}\left[1/b_{(x,y)}\right]$ over the randomness of all variables. Note that now that $X$ has not been observed, so the number of each block type in each set is an Poisson random variable, and it is independent from the rest. Let $F$ denote the set of all flattening blocks. We denote $p_1(x)\,k^{(p_1)}$, $p_2(y)\,k^{(p_2)}$, and $p(x,y)\,k^{(p)}$ by $\lambda_1$, $\lambda_2$, and $\lambda_3$ respectively. The expected value of $1/b_{(x,y)}$ can be rewritten as:

$$\mathbf{E}_F\left[\frac{1}{b_{(x,y)}}\right] = \mathbf{E}_F\left[\frac{1}{(k_x^{(p_1)}+1)(k_y^{(p_2)}+1)+k_{(x,y)}^{(p)}}\right]$$

$$\leq \mathbf{E}_F\left[\min\left(\frac{1}{(k_x^{(p_1)}+1)(k_y^{(p_2)}+1)},\frac{1}{k_{(x,y)}^{(p)}+1}\right)\right]$$

$$\leq \min\left(\mathbf{E}_F\left[\frac{1}{k_x^{(p_1)}+1}\right]\cdot\mathbf{E}_F\left[\frac{1}{k_y^{(p_2)}+1}\right],\mathbf{E}_F\left[\frac{1}{k_{(x,y)}^{(p)}+1}\right]\right)$$

$$\leq \min\left(\frac{1-e^{-\lambda_1}}{\lambda_1}\cdot\frac{1-e^{-\lambda_2}}{\lambda_2},\frac{1-e^{-\lambda_3}}{\lambda_3}\right) \leq \min\left(\frac{4}{(\lambda_1+1)(\lambda_2+1)},\frac{2}{\lambda_3+1}\right)$$

$$\leq \frac{8}{(\lambda_1+1)(\lambda_2+1)+\lambda_3}$$

$$= \frac{8}{(p_1(x)\,k^{(p_1)}+1)\cdot(p_2(y)\,k^{(p_2)}+1)+p(x,y)\,k^{(p)}}$$

Governed by the previous equations, we obtain:

$$\mathbf{Pr}_X\left[\mathbf{E}_\pi\left[\frac{1}{b_{(x,y)}(X,\pi)}\right] \geq \frac{1}{20000}\cdot\mathbf{E}_F\left[\frac{1}{b_{(x,y)}}\right]\right] \geq 0.96\,,$$

which is equivalent to

$$\mathbf{Pr}_X\left[\mathbf{E}_\pi\left[\frac{(p(x,y)-q(x,y))^2}{b_{(x,y)}(X,\pi)}\right] < \frac{1}{20000}\cdot\mathbf{E}_F\left[\frac{(p(x,y)-q(x,y))^2}{b_{(x,y)}}\right]\right] \leq 0.04\,.$$

To prove Equation 4, we need to show that the above equation holds even for the sum of the quantities over all $(x, y)$ with high probability. We show the claim in the following lemma. The proof is in Section D:

**Lemma C.4.** *Let $x_1, x_2, \ldots, x_n$ be $n$ non-negative random variables. Suppose there exist two constants $c$ and $p$, both at most one, such that for each random variable $x_i$, we have:*

$$\mathbf{Pr}[x_i < c\cdot\mathbf{E}[x_i]] \leq p\,,$$

*Then, one can show:*

$$\mathbf{Pr}\left[\sum_{i=1}^n x_i < \frac{c\cdot\sum_{i=1}^n\mathbf{E}[x_i]}{10}\right] \leq \frac{10\,p}{9}\,.$$

Now, we focus on proving Equation 5. To prove the inequality, it suffices to show that $\mathbf{E}_F\left[1/b_{(x,y)}^3\right]$ is $O\left(\mathbf{E}_F\left[1/b_{(x,y)}^2\right]^2\right)$. Again, note that we can think of $b_{(x,y)}$ to be equal to $(X+1)(Y+1) + Z$ where $X$, $Y$ and $Z$ are three Poisson random variables with means $\lambda_1 = p_1(x)k^{(p_1)}$, $\lambda_2 = p_2(y)k^{(p_2)}$, and $\lambda_3 =$ respectively. In Equation 12 we show that:

$$\mathbf{E}_{W\sim\mathbf{Poi}(\lambda)}\left[\frac{1}{W^3}\right] \leq 6 \cdot \left(\frac{1-e^{-\lambda}}{\lambda}\right)^2$$

735 Thus, we obtain an upper bound for the expected value of $1/b_{(x,y)}^3$ as follows:

$$\mathbf{E}_F\left[\frac{1}{b_{(x,y)}^3}\right] = \mathbf{E}_{X,Y,Z}\left[\frac{1}{((X+1)\cdot(Y+1)+Z)^3}\right] \leq \mathbf{E}_{X,Y,Z}\left[\min\left(\frac{1}{(X+1)^3}\cdot\frac{1}{(Y+1)^3}, \frac{1}{(Z+1)^3}\right)\right]$$

$$\leq \min\left(\mathbf{E}_X\left[\frac{1}{(X+1)^3}\right]\cdot\mathbf{E}_Y\left[\frac{1}{(Y+1)^3}\right], \mathbf{E}_Z\left[\frac{1}{(Z+1)^3}\right]\right)$$

$$\leq 36 \min\left(\left(\frac{1-e^{-\lambda_1}}{\lambda_1}\right)^2\cdot\left(\frac{1-e^{-\lambda_2}}{\lambda_2}\right)^2, \left(\frac{1-e^{-\lambda_3}}{\lambda_3}\right)^2\right).$$

736 Note that in the case that one of the $lambda$'s is equal to zero, one can replace $1 - e^{-\lambda}/\lambda$ by one
737 in the rest of the proof. On the other hand, we can find a lower bound for $1/b_{(x,y)}$ by Jensen's
738 inequality:

$$\left(\mathbf{E}_F\left[\frac{1}{b_{(x,y)}}\right]\right)^2 \geq \left(\frac{1}{\mathbf{E}_F[b_i]}\right)^2 = \left(\frac{1}{(\lambda_1+1)(\lambda_2+1)+\lambda_3}\right)^2$$

$$\geq \left(\frac{1}{2}\min\left(\frac{1}{\lambda_1+1}\cdot\frac{1}{\lambda_2+1}, \frac{1}{\lambda_3+1}\right)\right)^2$$

$$\geq \frac{1}{4}\min\left(\left(\frac{1}{\lambda_1+1}\cdot\frac{1}{\lambda_2+1}\right)^2, \left(\frac{1}{\lambda_3+1}\right)^2\right)$$

$$\geq \frac{1}{64}\min\left(\left(\frac{1-e^{-\lambda_1}}{\lambda_1}\cdot\frac{1-e^{-\lambda_2}}{\lambda_2}\right)^2, \left(\frac{1-e^{-\lambda_3}}{\lambda_3}\right)^2\right)$$

739 where the last inequality is due to the fact that $(1 - e^{-t})/t$ is at most $2/(t+1)$ for a non-negative
740 number $t$. Putting these two equations together, one can conclude Equation 5:

$$\mathbf{E}_F\left[\|p^{(F)} - q^{(F)}\|_4^4\right] = \sum_{x=1}^n\sum_{y=1}^m\mathbf{E}_F\left[\frac{(p(x,y)-q(x,y))^4}{b_{(x,y)}^3}\right] \leq 36\cdot64\cdot\sum_{x=1}^n\sum_{y=1}^m\left(\mathbf{E}_F\left[\frac{(p(x,y)-q(x,y))^2}{b_{x,y}}\right]\right)^2$$

$$\leq 2304\cdot\left(\sum_{x=1}^n\sum_{y=1}^m\mathbf{E}_F\left[\frac{(p(x,y)-q(x,y))^2}{b_{x,y}}\right]\right)^2 = 2304\cdot\mathbf{E}_F\left[\|p^{(F)}-q^{(F)}\|_2^2\right]^2.$$

741 Therefore, the statement of the lemma is concluded. □

742 3

### C.2 sensitivity of the statistic for the independence problem

744 In this section, we give an upper bound for the sensitivity of the independence statistic: the amount
745 that the statistic changes if we change one sample in the input.

Let $X$ denote a set of block that the algorithm received as the input. Assume we permute the blocks in $X$ using a permutation $\pi$. Note that if we fix the size of each flattening set and sample set, $\hat{s}_1, \hat{s}_2$, etc., one can deterministically find $s_{(x,y)}^{(p)}, s_{(x,y)}^{(q)}$, etc.. Thus, given $X$, $\pi$, and sizes of sets, one can compute the following statistic:

$$Z(X,\pi) := \sum_{x=1}^m\sum_{y=1}^n\frac{(s_{(x,y)}^{(p)} - s_{(x,y)}^{(q)})^2 - s_{(x,y)}^{(p)} - s_{(x,y)}^{(q)}}{(k_x^{(p_1)}+1)(k_y^{(p_2)}+1) + k_{(x,y)}^{(p)} + k_{(x,y)}^{(q)}}$$

We denote the average of $Z$ over all $\pi$ by $\overline{Z}(X)$. Our goal here is to calculate

$$\Delta Z = \max X, X' |\overline{Z}(X) - \overline{Z}(X')|$$

where $X$ and $X'$ are two neighboring data sets that they differ in exactly one element.

Through this section, we use an important property of Poissonization method: Let $A$ and $B$ be two sets with $\hat{n}_1 = \mathbf{Poi}(n_1)$ and $\hat{n}_2 = \mathbf{Poi}(n_2)$ samples. Given that there are $k$ instance of element $i$ in $A$ and $B$ together, the number of instances of element $i$ in $A$ is a Binomial random variable: $\mathbf{Bin}(\hat{n}_1 + \hat{n}_2, n_1/(n_1 + n_2))$.

**Lemma C.3.** *Given that the size of all flattening and test samples are within the constant factor of their expectations, the sensitivity of the statistic $Z$ is bounded as follows:*

$$\Theta\left( \frac{s}{k^{(q)}} + \frac{s}{k^{(p)}} + \frac{s}{k^{(p)}} \cdot \frac{f_{\langle(.,b),(.,.)\rangle}}{f_{\langle(.,.),(.,b)\rangle} + 1} \right)$$

**Proof:** In this proof, we assume $X$ and $X'$ are fully given, and $X$ and $X'$ are only different in the $r$-th block of the samples. Note that when we permute the elements in $X$ and $X'$, we only permute the blocks and do not change the order of the samples within each block. The expectations in the this proof are taken over the random choice of a permutation $\pi$. As we mentioned earlier, we partition the blocks into the following sets, and the number of occurrences of each block types in each set determines the statistic:

$$\mathcal{S} = \left\{ F^{(p_1)}, F^{(p_2)}, F^{(p)}, F^{(q)}, T^{(p)}, T^{(q)} \right\}$$

We can separate our calculation based on where the $r$-th block is:

$$\begin{aligned} \Delta(\overline{Z}) &= |\overline{Z}(X) - \overline{Z}(X')| \\ &= \left| \mathbf{E}_\pi[Z(X, \pi) - Z(X', \pi)] \right| \leq \mathbf{E}_\pi[|Z(X, \pi) - Z(X', \pi)|] \\ &= \sum_{S \in \mathcal{S}} \mathbf{Pr}_\pi[r \in S] \cdot \left| \mathbf{E}_\pi[|Z(X, \pi) - Z(X', \pi)| \,|\, r \in S] \right| \end{aligned}$$

Now, we consider each term separate.

1. **Block $r$ is in $F^{(p_1)}$:** Suppose the types of the $r$-th block in $X$ and $X'$ are $\langle(a, .), (., .)\rangle$ and $\langle(a', .), (., .)\rangle$ respectively. If $a$ and $a'$ are equal, then the statistic will remains unchanged. Otherwise, $k_a^{(p_1)}$ and $k_{a'}^{\prime (p_1)}$ is changed by one. First, we simplify the term $|Z(X, \pi) - Z(X', \pi)|$ for a given $\pi$:

$$\begin{aligned} |Z(X, \pi) - Z(X', \pi)| &= \left| \sum_{x=1}^{n} \sum_{y=1}^{m} \frac{(s_{(x,y)}^{(p)} - s_{(x,y)}^{(q)})^2 - s_{(x,y)}^{(p)} - s_{(x,y)}^{(q)}}{(k_x^{(p_1)} + 1)(k_y^{(p_2)} + 1) + k_{(x,y)}^{(p)} + k_{(x,y)}^{(q)}} \right. \\ &\quad \left. - \sum_{x=1}^{n} \sum_{y=1}^{m} \frac{(s_{(x,y)}^{\prime(p)} - s_{(x,y)}^{\prime(q)})^2 - s_{(x,y)}^{\prime(p)} - s_{(x,y)}^{\prime(q)}}{(k_x^{\prime(p_1)} + 1)(k_y^{\prime(p_2)} + 1) + k_{(x,y)}^{\prime(p)} + k_{(x,y)}^{\prime(q)}} \right| \\ &\leq \sum_{x \in \{a, a'\}} \left| \sum_{y=1}^{m} \frac{(s_{(x,y)}^{(p)} - s_{(x,y)}^{(q)})^2 - s_{(x,y)}^{(p)} - s_{(x,y)}^{(q)}}{(k_x^{(p_1)} + 1)(k_y^{(p_2)} + 1) + k_{(x,y)}^{(p)} + k_{(x,y)}^{(q)}} \right. \\ &\quad \left. - \frac{(s_{(x,y)}^{\prime(p)} - s_{(x,y)}^{\prime(q)})^2 - s_{(x,y)}^{\prime(p)} - s_{(x,y)}^{\prime(q)}}{(k_x^{\prime(p_1)} + 1)(k_y^{\prime(p_2)} + 1) + k_{(x,y)}^{\prime(p)} + k_{(x,y)}^{\prime(q)}} \right| \end{aligned}$$

For the rest of the proof, we focus on the term above when $x = a$. The other term can be upper bounded similarly, and at the end we multiply our final bound by two.

$$\left| \sum_{y=1}^{m} \frac{(s_{(a,y)}^{(p)} - s_{(a,y)}^{(q)})^2 - s_{(a,y)}^{(p)} - s_{(a,y)}^{(q)}}{(k_a^{(p_1)} + 1)(k_y^{(p_2)} + 1) + k_{(a,y)}^{(p)} + k_{(x,y)}^{(q)}} - \frac{(s'_{(a,y)}^{(p)} - s'_{(a,y)}^{(q)})^2 - s'_{(a,y)}^{(p)} - s'_{(a,y)}^{(q)}}{(k'_a(p_1) + 1)(k'_y^{(p_2)} + 1) + k'_{(a,y)}^{(p)} + k'_{(x,y)}^{(q)}} \right|$$

$$\leq \sum_{y=1}^{m} \left| \frac{(s_{(a,y)}^{(p)} - s_{(a,y)}^{(q)})^2 - s_{(a,y)}^{(p)} - s_{(a,y)}^{(q)}}{(k'_a^{(p_1)} + 2)(k_y^{(p_2)} + 1) + k_{(a,y)}^{(p)} + k_{(x,y)}^{(q)}} - \frac{(s_{(a,y)}^{(p)} - s_{(a,y)}^{(q)})^2 - s_{(a,y)}^{(p)} - s_{(a,y)}^{(q)}}{(k'_a(p_1) + 1)(k_y^{(p_2)} + 1) + k_{(a,y)}^{(p)} + k_{(x,y)}^{(q)}} \right|$$

$$\leq \sum_{y=1}^{m} \frac{\left( k_y^{(p_2)} + 1 \right) \cdot \left| (s_{(a,y)}^{(p)} - s_{(a,y)}^{(q)})^2 - s_{(a,y)}^{(p)} - s_{(a,y)}^{(q)} \right|}{\left( (k'_a^{(p_1)} + 2)(k_y^{(p_2)} + 1) + k_{(a,y)}^{(p)} + k_{(x,y)}^{(q)} \right) \cdot \left( (k'_a(p_1) + 1)(k_y^{(p_2)} + 1) + k_{(a,y)}^{(p)} + k_{(x,y)}^{(q)} \right)}$$

$$\leq \sum_{y=1}^{m} \frac{\left( k_y^{(p_2)} + 1 \right) \cdot \left( (s_{(a,y)}^{(p)})^2 + (s_{(a,y)}^{(q)})^2 \right)}{\left( (k'_a^{(p_1)} + 2)(k_y^{(p_2)} + 1) + k_{(a,y)}^{(p)} + k_{(x,y)}^{(q)} \right) \cdot \left( (k'_a(p_1) + 1)(k_y^{(p_2)} + 1) + k_{(a,y)}^{(p)} + k_{(x,y)}^{(q)} \right)}$$

$$\leq \sum_{y=1}^{m} \frac{1}{k_a^{(p_1)} + 1} \cdot \frac{(s_{(a,y)}^{(p)})^2}{k_{(a,y)}^{(p)} + 1} + \frac{1}{k_a^{(p_1)} + 1} \cdot \frac{(s_{(a,y)}^{(q)})^2}{k_{(a,y)}^{(q)} + 1}$$

For brevity's sake, let $v$ denote the following expectation:

$$v := \mathbf{E}_\pi \left[ \sum_{y=1}^{m} \frac{1}{k_a^{(p_1)} + 1} \cdot \frac{(s_{(a,y)}^{(p)})^2}{k_{(a,y)}^{(p)} + 1} + \frac{1}{k_a^{(p_1)} + 1} \cdot \frac{(s_{(a,y)}^{(q)})^2}{k_{(a,y)}^{(q)} + 1} \middle| r \in F^{(p_1)} \right]$$

Using the tower rule, we achieve:

$$v \leq \mathbf{E}_{F^{(p_1)}} \left[ \frac{1}{(k_a^{(p_1)} + 1)} \cdot \mathbf{E}_\pi \left[ \sum_{y=1}^{m} \frac{(s_{(a,y)}^{(p)})^2}{(k_{(a,y)}^{(p)} + 1)} \middle| r \in F^{(p_1)}, F^{(p_1)} \right] \middle| r \in F^{(p_1)} \right]$$

$$+ \mathbf{E}_{F^{(p_1)}} \left[ \frac{1}{k_a^{(p_1)} + 1} \cdot \mathbf{E}_\pi \left[ \sum_{y=1}^{m} \frac{(s_{(a,y)}^{(q)})^2}{k_{(a,y)}^{(q)} + 1} \middle| r \in F^{(p_1)}, F^{(p_1)} \right] \middle| r \in F^{(p_1)} \right]$$

Let $f_{\langle(a,y),(.,.)\rangle}$, $f_{\langle(a,.),(.,y)\rangle}$, and $f_{\langle(a,.),(.,.)\rangle}$ be the numbers of blocks of the forms $\langle(a,y),(.,.)\rangle$, $\langle(a,.),(.,y)\rangle$, and $\langle(a,.),(.,.)\rangle$ in $X$ respectively. Using Lemma D.4, one can bound the terms inside the expectations as below:

$$\mathbf{E}\left[ \frac{(s_{(a,y)}^{(p)})^2}{(k_{(a,y)}^{(p)} + 1)} \right] \leq \min\left( \frac{2(s-1) f_{\langle(a,y),(.,.)\rangle}}{(k^{(p)} + 1)}, 2 f_{\langle(a,y),(.,.)\rangle}^2 \right) + f_{\langle(a,y),(.,.)\rangle}$$

$$\leq \min\left( \left( \frac{2(s-1)}{(k^{(p)} + 1)} + 1 \right) \cdot f_{\langle(a,y),(.,.)\rangle}, 3 f_{\langle(a,y),(.,.)\rangle}^2 \right),$$

$$\mathbf{E}\left[ \frac{(s_{(a,y)}^{(q)})^2}{(k_{(a,y)}^{(q)} + 1)} \right] \leq \min\left( \frac{2(s-1) f_{\langle(a,.),(.,y)\rangle}}{(k^{(q)} + 1)}, 2 f_{\langle(a,.),(.,y)\rangle}^2 \right) + f_{\langle(a,.),(.,y)\rangle}$$

$$\leq \min\left( \left( \frac{2(s-1)}{(k^{(q)} + 1)} + 1 \right) \cdot f_{\langle(a,.),(.,y)\rangle}, 3 f_{\langle(a,.),(.,y)\rangle}^2 \right).$$

Observe that $\sum_y f_{\langle (a,y),(.,.)\rangle}$ and $\sum_y f_{\langle (a,.),(.,y)\rangle}$ are equal to $f_{\langle (a,.),(.,.)\rangle}$. Thus, using Lemma D.5, and Lemma D.6, we have:

$$v \leq \mathbf{E}_{F^{(p_1)}}\left[\left.\frac{1}{(k_a^{(p_1)}+1)}\right| r \in F^{(p_1)}\right] \cdot \sum_{y=1}^m \min\left(\left(\frac{2(s-1)}{(k^{(p)}+1)}+1\right) \cdot f_{\langle (a,y),(.,.)\rangle}, 3\, f^2_{\langle (a,y),(.,.)\rangle}\right)$$

$$+ \mathbf{E}_{F^{(p_1)}}\left[\left.\frac{1}{k_a^{(p_1)}+1}\right| r \in F^{(p_1)}\right] \cdot \sum_{y=1}^m \min\left(\left(\frac{2(s-1)}{(k^{(q)}+1)}+1\right) \cdot f_{\langle (a,.),(.,y)\rangle}, 3\, f^2_{\langle (a,.),(.,y)\rangle}\right)$$

$$\leq \min\left(1, \frac{|X|}{f_{\langle (a,.),(.,.)\rangle}\, k^{(p_1)}}\right) \cdot \left(\frac{2(s-1)}{(k^{(p)}+1)}+1\right) \cdot f_{\langle (a,.),(.,.)\rangle}$$

$$+ \min\left(1, \frac{|X|}{f_{\langle (a,.),(.,.)\rangle}\, k^{(p_1)}}\right) \cdot \left(\frac{2(s-1)}{(k^{(q)}+1)}+1\right) \cdot f_{\langle (a,.),(.,.)\rangle}$$

$$\leq \Theta\left(\frac{|X|}{k^{(p_1)}} \cdot \left(\frac{s}{k^{(p)}} + \frac{s}{k^{(q)}}\right)\right)$$

Using the above calculation, it is not hard to see that the following holds

$$\mathbf{Pr}_\pi\left[r \in F^{(p_1)}\right] \cdot \left|\mathbf{E}_\pi\left[|Z(X,\pi) - Z(X',\pi)|\,|\, r \in F^{(p_1)}\right]\right|$$

$$\leq 2 \cdot \frac{k^{(p_1)}}{|X|} \cdot \left(\frac{|X|}{k^{(p_1)}} \cdot \left(\frac{s}{k^{(p)}} + \frac{s}{k^{(q)}}\right)\right)$$

$$\leq \Theta\left(\frac{s}{k^{(p)}} + \frac{s}{k^{(q)}}\right).$$

2. **Block $r$ is in $F^{(p_2)}$:** Suppose the the $r$-th block in $X$ and $X'$ are of the forms $\langle (.,.),(.,b)\rangle$ and $\langle (.,.),(.,b')\rangle$ respectively. Using the symmetry of this case and the previous case, we take the same approach.

$$\mathbf{E}_\pi\left[\left.\sum_{x=1}^n \frac{1}{k_b^{(p_2)}+1} \cdot \frac{(s_{(x,b)}^{(p)})^2}{k_{(x,b)}^{(p)}+1} + \frac{1}{k_b^{(p_2)}+1} \cdot \frac{(s_{(x,b)}^{(q)})^2}{k_{(x,b)}^{(q)}+1}\right| r \in F^{(p_2)}\right]$$

$$\leq \mathbf{E}_{F^{(p_2)}}\left[\left.\frac{1}{k_b^{(p_2)}+1} \cdot \mathbf{E}_\pi\left[\left.\sum_{x=1}^n \frac{(s_{(x,b)}^{(p)})^2}{k_{(x,b)}^{(p)}+1}\right| r \in F^{(p_2)}, F^{(p_2)}\right]\right| r \in F^{(p_2)}\right]$$

$$+ \mathbf{E}_{F^{(p_2)}}\left[\left.\frac{1}{k_b^{(p_2)}+1} \cdot \mathbf{E}_\pi\left[\left.\sum_{x=1}^n \frac{(s_{(x,b)}^{(q)})^2}{k_{(x,b)}^{(q)}+1}\right| r \in F^{(p_2)}, F^{(p_2)}\right]\right| r \in F^{(p_2)}\right]$$

$$\leq \mathbf{E}_{F^{(p_2)}}\left[\left.\frac{1}{k_b^{(p_2)}+1}\right| r \in F^{(p_2)}\right] \cdot \sum_{x=1}^n \min\left(\left(\frac{2(s-1)}{(k^{(p)}+1)}+1\right) \cdot f_{\langle (x,b),(.,.)\rangle}, 3\, f^2_{\langle (x,b),(.,.)\rangle}\right)$$

$$+ \mathbf{E}_{F^{(p_2)}}\left[\left.\frac{1}{k_b^{(p_2)}+1}\right| r \in F^{(p_2)}\right] \cdot \sum_{x=1}^n \min\left(\left(\frac{2(s-1)}{(k^{(q)}+1)}+1\right) \cdot f_{\langle (x,.),(.,b)\rangle}, 3\, f^2_{\langle (x,.),(.,b)\rangle}\right)$$

$$\leq \min\left(1, \frac{|X|}{f_{\langle (.,.),(.,b)\rangle}\, k^{(p_2)}}\right) \cdot \left(\frac{2(s-1)}{(k^{(p)}+1)}+1\right) \cdot f_{\langle (.,b),(.,.)\rangle}$$

$$+ \min\left(1, \frac{|X|}{f_{\langle (.,.),(.,b)\rangle}\, k^{(p_2)}}\right) \cdot \left(\frac{2(s-1)}{(k^{(q)}+1)}+1\right) \cdot f_{\langle (.,.),(.,b)\rangle}$$

$$\leq \frac{|X|}{k^{(p_2)}} \cdot \left(\frac{2(s-1)}{(k^{(p)}+1)}+1\right) \cdot \frac{f_{\langle (.,b),(.,.)\rangle}}{f_{\langle (.,.),(.,b)\rangle}+1} + \frac{|X|}{k^{(p_2)}} \cdot \left(\frac{2(s-1)}{(k^{(q)}+1)}+1\right)$$

$$\leq \Theta\left(\frac{|X|}{k^{(p_2)}} \cdot \left(\frac{s}{k^{(p)}} \cdot \frac{f_{\langle (.,b),(.,.)\rangle}}{f_{\langle (.,.),(.,b)\rangle}+1} + \frac{s}{k^{(q)}}\right)\right)$$

where the last inequality is due to the fact that $\min_{x>0}(\alpha/x, x) \leq \sqrt{\alpha}$.

$$\mathbf{Pr}_\pi\left[r \in F^{(p_2)}\right] \cdot \left|\mathbf{E}_\pi\left[\,|Z(X,\pi) - Z(X',\pi)|\,\big|\,r \in F^{(p_2)}\right]\right|$$

$$\leq 2 \cdot \frac{k^{(p_2)}}{|X|} \cdot \Theta\left(\frac{|X|}{k^{(p_2)}} \cdot \left(\frac{s}{k^{(p)}} \cdot \frac{f_{\langle(.,b),(.,.)\rangle}}{f_{\langle(.,.),(.,b)\rangle} + 1} + \frac{s}{k^{(q)}}\right)\right)$$

$$\leq \Theta\left(\frac{s}{k^{(p)}} \cdot \frac{f_{\langle(.,b),(.,.)\rangle}}{f_{\langle(.,.),(.,b)\rangle} + 1} + \frac{s}{k^{(q)}}\right)$$

3. **Block $r$ is in $F^{(p)}$ or $F^{(q)}$ :** Here we assume $r$ is in $F^{(p)}$. Very similar calculation, yield the same bound if $r$ is in $F^{(q)}$. Suppose the the $r$-th block in $X$ and $X'$ are of the forms $\langle(a,b),(.,.)\rangle$ and $\langle(a',b'),(.,.)\rangle$ respectively. Note that in this case, only two terms will be different, so we have:

$$|Z(X,\pi) - Z(X',\pi)| \leq \sum_{\substack{(x,y)\in \\ \{(a,b),(a',b')\}}} \left| \frac{(s^{(p)}_{(x,y)} - s^{(q)}_{(x,y)})^2 - s^{(p)}_{(x,y)} - s^{(q)}_{(x,y)}}{(k^{(p_1)}_x + 1)(k^{(p_2)}_y + 1) + k^{(p)}_{(x,y)} + k^{(q)}_{(x,y)}} \right.$$

$$\left. - \frac{(s'^{(p)}_{(x,y)} - s'^{(q)}_{(x,y)})^2 - s'^{(p)}_{(x,y)} - s'^{(q)}_{(x,y)}}{(k'^{(p_1)}_x + 1)(k'^{(p_2)}_y + 1) + k'^{(p)}_{(x,y)} + k^{(q)}_{(x,y)}} \right|$$

Now, we focus on the term where $(x,y)$ is equal to $(a,b)$. The other term can be bounded similarly.

$$\left| \frac{(s^{(p)}_{(a,b)} - s^{(q)}_{(a,b)})^2 - s^{(p)}_{(a,b)} - s^{(q)}_{(a,b)}}{(k^{(p_1)}_a + 1)(k^{(p_2)}_b + 1) + k^{(p)}_{(a,b)} + k^{(q)}_{(a,b)}} - \frac{(s'^{(p)}_{(a,b)} - s'^{(q)}_{(a,b)})^2 - s'^{(p)}_{(a,b)} - s'^{(q)}_{(a,b)}}{(k'^{(p_1)}_a + 1)(k'^{(p_2)}_b + 1) + k'^{(p)}_{(a,b)} + k'^{(q)}_{(a,b)}} \right|$$

$$= \left| \frac{(s^{(p)}_{(a,b)} - s^{(q)}_{(a,b)})^2 - s^{(p)}_{(a,b)} - s^{(q)}_{(a,b)}}{(k^{(p_1)}_a + 1)(k^{(p_2)}_b + 1) + k^{(p)}_{(a,b)} + k^{(q)}_{(a,b)}} - \frac{(s^{(p)}_{(a,b)} - s^{(q)}_{(a,b)})^2 - s^{(p)}_{(a,b)} - s^{(q)}_{(a,b)}}{(k^{(p_1)}_a + 1)(k^{(p_2)}_b + 1) + k^{(p)}_{(a,b)} + k^{(q)}_{(a,b)} + 1} \right|$$

$$\leq \frac{(s^{(p)}_{(a,b)})^2 + (s^{(q)}_{(a,b)})^2}{\left((k^{(p_1)}_a + 1)(k^{(p_2)}_b + 1) + k^{(p)}_{(a,b)} + k^{(q)}_{(a,b)}\right) \cdot \left((k^{(p_1)}_a + 1)(k^{(p_2)}_b + 1) + k^{(p)}_{(a,b)} + k^{(q)}_{(a,b)} + 1\right)}$$

$$\leq \frac{(s^{(p)}_{(a,b)})^2}{k^{(p)}_{(a,b)}(k^{(p)}_{(a,b)} + 1)} + \frac{(s^{(q)}_{(a,b)})^2}{k^{(q)}_a(k^{(q)}_a + 1)}$$

Now, using Lemma D.6, we bound the expected value of the above quantity from above:

$$\mathbf{E}_\pi\left[ \frac{(s^{(p)}_{(a,b)})^2}{k^{(p)}_{(a,b)}(k^{(p)}_{(a,b)} + 1)} + \frac{(s^{(q)}_{(a,b)})^2}{k^{(q)}_a(k^{(q)}_a + 1)} \,\bigg|\, r \in F^{(p)} \right]$$

$$\leq f^2_{\langle(a,b),(.,.)\rangle} \cdot \mathbf{E}_\pi\left[ \frac{1}{k^{(p)}_{(a,b)}(k^{(p)}_{(a,b)} + 1)} \,\bigg|\, r \in F^{(p)} \right]$$

$$+ f^2_{\langle(a,.),(.,b)\rangle} \cdot \mathbf{E}_\pi\left[ \frac{1}{k^{(q)}_a(k^{(q)}_a + 1)} \,\bigg|\, r \in F^{(p)} \right]$$

$$\leq \frac{|X|(|X| + 1)}{k^{(p)}(k^{(p)} + 1)} + \frac{|X|(|X| + 1)}{k^{(q)}(k^{(q)} + 1)}$$

Using the above equation, and the fact that $|X| = \Theta(s)$, it is not hard to see that

$$\mathbf{Pr}_\pi\Big[r \in F^{(p)}\Big] \cdot \Big|\mathbf{E}_\pi\Big[\,|Z(X,\pi) - Z(X',\pi)|\,\big|\,r \in F^{(p)}\Big]\Big|$$

$$\leq 2 \cdot \frac{k^{(p)}}{|X|} \cdot \Theta\left(\frac{|X|(|X|+1)}{k^{(p)}\,(k^{(p)}+1)} + \frac{|X|(|X|+1)}{k^{(q)}\,(k^{(q)}+1)}\right)$$

$$\leq \Theta\left(\frac{s}{k^{(p)}} + \frac{s}{k^{(q)}}\right)$$

Note that the factor of two in the above inequality, comes from including the symmetric term for $(a', b')$.

4. **Block $r$ is in $T^{(p)}$ or $T^{(q)}$:** Suppose the the $r$-th block in $X$ and $X'$ are of the forms $\langle (a,b), (.,.)\rangle$ and $\langle (a',b'), (.,.)\rangle$ respectively. Note that in this case, only two terms will be different for the two datasets, so we have:

$$|Z(X,\pi) - Z(X',\pi)| \leq \sum_{\substack{(x,y)\in \\ \{(a,b),(a',b')\}}} \left| \frac{(s^{(p)}_{(x,y)} - s^{(q)}_{(x,y)})^2 - s^{(p)}_{(x,y)} - s^{(q)}_{(x,y)}}{(k^{(p_1)}_x + 1)(k^{(p_2)}_y + 1) + k^{(p)}_{(x,y)} + k^{(q)}_{(x,y)}} \right.$$

$$\left. - \frac{(s'^{(p)}_{(x,y)} - s'^{(q)}_{(x,y)})^2 - s'^{(p)}_{(x,y)} - s'^{(q)}_{(x,y)}}{(k'^{(p_1)}_x + 1)(k'^{(p_2)}_y + 1) + k'^{(p)}_{(x,y)} + k'^{(q)}_{(x,y)}} \right|$$

Below, we assume $r$ is in $T^{(p)}$. However, the calculation will be the same if $r$ was in $T^{(q)}$. Now, we focus on the term where $(x,y)$ is equal to $(a,b)$. The other term can be bounded similarly.

$$\left| \frac{(s^{(p)}_{(a,b)} - s^{(q)}_{(a,b)})^2 - s^{(p)}_{(a,b)} - s^{(q)}_{(a,b)}}{(k^{(p_1)}_a + 1)(k^{(p_2)}_b + 1) + k^{(p)}_{(a,b)} + k^{(q)}_{(a,b)}} - \frac{(s'^{(p)}_{(a,b)} - s'^{(q)}_{(a,b)})^2 - s'^{(p)}_{(a,b)} - s'^{(q)}_{(a,b)}}{(k'^{(p_1)}_a + 1)(k'^{(p_2)}_b + 1) + k'^{(p)}_{(a,b)} + k'^{(q)}_{(a,b)}} \right|$$

$$= \left| \frac{(s^{(p)}_{(a,b)} - s^{(q)}_{(a,b)})^2 - s^{(p)}_{(a,b)} - s^{(q)}_{(a,b)}}{(k^{(p_1)}_a + 1)(k^{(p_2)}_b + 1) + k^{(p)}_{(a,b)} + k^{(q)}_{(a,b)}} - \frac{(s^{(p)}_{(a,b)} - 1 - s^{(q)}_{(a,b)})^2 - (s^{(p)}_{(a,b)} - 1) - s^{(q)}_{(a,b)}}{(k^{(p_1)}_a + 1)(k^{(p_2)}_b + 1) + k^{(p)}_{(a,b)} + k^{(q)}_{(a,b)}} \right|$$

$$\leq \frac{2s^{(p)}_{(a,b)} + 2s^{(q)}_{(a,b)}}{(k^{(p_1)}_a + 1)(k^{(p_2)}_b + 1) + k^{(p)}_{(a,b)} + k^{(q)}_{(a,b)}} \leq \frac{2s^{(p)}_{(a,b)}}{k^{(p)}_{(a,b)} + 1} + \frac{2s^{(q)}_{(a,b)}}{k^{(q)}_a + 1}$$

Now, using Lemma D.5 , we bound the expected value of the above quantity from above:

$$\mathbf{E}_\pi\left[ \frac{2s^{(p)}_{(a,b)}}{k^{(p)}_{(a,b)} + 1} + \frac{2s^{(q)}_{(a,b)}}{k^{(q)}_a + 1} \,\middle|\, r \in F^{(p)} \right]$$

$$\leq 2\,f_{\langle(a,b),(.,.)\rangle} \cdot \mathbf{E}_\pi\left[ \frac{1}{k^{(p)}_{(a,b)} + 1} \,\middle|\, r \in F^{(p)} \right]$$

$$+ 2\,f_{\langle(a,.),(.,b)\rangle} \cdot \mathbf{E}_\pi\left[ \frac{1}{k^{(q)}_a + 1} \,\middle|\, r \in F^{(p)} \right]$$

$$\leq \frac{|X|}{k^{(p)} + 1} + \frac{|X|}{k^{(q)} + 1}$$

Using the above equation, and the fact that $|X| = \Theta(s)$, it is not hard to see that

$$\mathbf{Pr}_\pi\Big[r \in F^{(p)}\Big] \cdot \Big|\mathbf{E}_\pi\Big[\,|Z(X,\pi) - Z(X',\pi)|\,\big|\,r \in F^{(p)}\Big]\Big|$$

$$\leq 2 \cdot \frac{k^{(p)}}{|X|} \cdot \Theta\left(\frac{|X|}{k^{(p)} + 1} + \frac{|X|}{k^{(q)} + 1}\right) \leq \Theta(1)$$

Note that the factor of two in the above inequality, comes from including the symmetric term for $(a', b')$.

Putting all the terms computed above together, we obtain:

$$|\overline{Z}(X) - \overline{Z}(X')| = \sum_{S \in \mathcal{S}} \mathbf{Pr}_\pi[r \in S] \cdot \left| \mathbf{E}_\pi[\, |Z(X, \pi) - Z(X', \pi)| \,|\, r \in S] \right|$$

$$\leq \Theta\left( \frac{s}{k^{(q)}} + \frac{s}{k^{(p)}} + \frac{s}{k^{(p)}} \cdot \frac{f_{\langle (.,b),(.,.)\rangle}}{f_{\langle (.,.),(.,b)\rangle} + 1} \right)$$

□

## C.3  Stretching the domain of a private algorithm

In this section, we investigate whether we can extend the domain of a differentially private tester under certain conditions. We start off by defining the domains. The input of a differential private tester is a sample set from a universe $\Omega$. Suppose we have a dataset $X$ of $2s$ samples from $[n]$ that are arranged in two rows each of size $s$ (namely top and bottom rows). Let the domain of a differential algorithm, denoted by $\mathcal{X}$, be the set of all such pairs of rows, namely $[n]^{2s}$. We denote the frequency of an element $i \in [n]$ in the top row by $t_i(X)$, and in the bottom row by $b_i(X)$. A desired property of a dataset is that the ratio of the frequencies in the row is bounded by a fixed parameter $A \geq 2$. More precisely, we define the subset of $\mathcal{X}$ which contains the data sets with this property as:

$$\mathcal{X}^* = \left\{ X \,\middle|\, \forall i \in [n] : \frac{t_i(X)}{b_i(X) + 1} \leq A \right\}$$

Let $\mathcal{A}$ be a tester that receives a set of samples, $X$, as its input, and outputs $\mathcal{A}(X)$ which is known to be correct with probability at least $1 - \delta$. Suppose $\mathcal{A}$ is $\xi$-differentially private when $X$ is in $\mathcal{X}^*$. Our goal here is to design a $\Theta(\xi)$-differentially private algorithm, namely $\mathcal{B}$, that takes $X \in \mathcal{X}$ as its input, and outputs $\mathcal{B}(X)$ which is incorrect with probability slightly larger than $\delta$.

At a high level, we implement $\mathcal{B}$ by using $\mathcal{A}$ as a blackbox as follows: We first look at the input $X \in \mathcal{X}$. If $X$ is already in $\mathcal{X}^*$, we output $\mathcal{A}(X)$. Otherwise, if $X$ is in $\mathcal{X} \setminus \mathcal{X}^*$, we pass $X$ through a "filter" and turn it into another dataset $Y$ which is in $\mathcal{X}^*$. Then, we output $\mathcal{A}(Y)$.

To show that $\mathcal{B}$ is the desired algorithm, we have few challenges: (1) we need to show the mapping does not affect the correctness probability by too much. (2) $\mathcal{B}$ is $\Theta(\xi)$-differentially private although its input may be from $\mathcal{X} \setminus \mathcal{X}^*$. Overcoming the second challenge is closely related to the design of the mapping. If two datasets have Hamming distance one, then we need to make sure they will remain "close". In the following section, we explain the mapping, and in the next section, we prove that $\mathcal{B}$ is a $\xi$-differentially private algorithm with large correctness probability.

## C.4  Mapping datasets in $\mathcal{X} \setminus \mathcal{X}^*$ to datasets in $\mathcal{X}^*$

In this section, we provide a randomized mapping that takes $X \in \mathcal{X}$ as the input, and maps it to randomly selected $Y$ in $\mathcal{X}^*$ with two important properties stated in the following Lemma:

**Lemma C.5.** *There exists a randomized mapping that takes $X, X' \in \mathcal{X}$ and maps them to $Y, Y' \in \mathcal{X}^*$ respectively with the following property:*

- *If $X$ is in $\mathcal{X}^*$, then it will always be mapped to itself. : $Y = X$.*

- *If the Hamming distance between $X$ and $X'$ is one, then there exists a coupling $\mathcal{C}$ between the random outputs of the mapping, $Y$ and $Y'$, where for any $(Y, Y')$ drawn from $\mathcal{C}$, the Hamming distance between $Y$ and $Y'$ is at most a constant $c = 4$ (independent of $A$).*

**Proof:** The main idea is to decrease the ratio $t_i(X)/(b_i(X) + 1)$ by replacing a subset of samples in the bottom row with the copies of $i$ to decrease the ratio without introducing new elements that violate the ratio condition. For a dataset $X$, we look at each element $i \in [n]$, and see how many copies of $i$ are needed to "fix" the ratio. It is not hard to see that if for each element $i$, $r_i(X)$ many copies is sufficient where $r_i(X)$ is defined as below:

$$r_i(X) := \max\left( \left\lceil \frac{t_i(X)}{A} \right\rceil - b_i(X) - 1, 0 \right).$$

Let $R$ be a multiset that contains $r_i(X)$ copies of $i$. We find $|R|$ slots in the bottom row, and replace the samples in those slots with an element in $R$. If we carefully select the slots and do not replace any copy of $i$ in the bottom row, the new ratio will be: $t_i(X)/(b_i(X) + r_i(X) + 1)$ which is at most $A$. Now, we focus on finding the slots in the bottom row. We can select a slot containing an instance of $i$, only if the replacement of $i$ does not increase the ratio of the frequencies above $A$. For an element $i$, we may remove at most $s_i(X)$ samples where

$$s_i(X) := \max\left(b_i(X) + 1 - \left\lceil \frac{t_i(X)}{A} \right\rceil, 0\right).$$

For each element $i$, we mark $s_i(X)$ many slots which contains copy of $i$ in the bottom row as "available" preferring the slots with the smaller index. Observe that we always have at least $|R|$ many slots since $A$ is at least two:

$$|R| = \sum_{i=1}^{n} r_i(X) \leq \sum_{i=1}^{n} \frac{t_i(X)}{A} \leq \frac{s}{A} \leq (A-1)\frac{s}{A} = s - \sum_{i=1}^{n} \frac{t_i(X)}{A} \leq s - \sum_{i=1}^{n}\left(\left\lceil \frac{t_i(X)}{A} \right\rceil - 1\right)$$

$$\leq s - \sum_{i=1}^{n} b_i(X) - s_i(X) = \sum_{i=1}^{n} s_i(X)$$

We choose the first $|R|$ available slots (i.e. with the smaller indices), and replace the bottom samples in them by the samples in $R$ randomly. After the replacements, it is clear that we did not remove a sample where its ratio could go above $A$, and we fixed all those elements with the ratio above $A$ as well. Thus, the dataset we get after this process is surely in $\mathcal{X}^*$. Furthermore, if $X$ is already in $\mathcal{X}^*$, then $R$ is an empty set, and the mapping does not change it, so $Y = X$.

Now, we focus on the proof of the existence of the coupling. Let $S$ be the indices of the $|R|$ available slots we select. First note that we consider all the elements in $R$ to be distinct. (even though they might be different copies of the same sample, we can index $r_i(X)$ copies of $i$ by $1, 2, \ldots, r_i(X)$.) Thus, there are $|R|!$ for assigning the samples in $R$ into the slots in $S$, and each assigning has probability $1/|R|!$. Suppose two datasets, $X$ and $X'$, differ in exactly one sample: $X$ has an extra copy of $i$, and $X'$ instead has an extra copy of $j$. Also, let $R'$ and $S'$ be the equivalents of $R$ and $S$ respectively for $X'$. Clearly, we have $|R| = |S|$, and $|R'| = |S'|$. This discrepancy between $X$ and $X'$ happens in either on the top row or the bottom row. Since the frequency of $i$ and $j$ changes by at most one, $r_i(X)$, $s_i(X)$, $r_j(X)$, and $s_j(X)$ will change by at most. Without loss of generality, if we consider all possible cases, it is not hard to see that one of the two following cases happens:

**Case 1:** $R$ and $R'$ has the same size, and $|R \cap R'|$ and $|S \cap S'|$ is at least $|R| - 1$. It is not hard to see that there is a bijection between $Y$ and $Y'$. Assume there exists a set of replacement that turns $X$ into $Y$. We construct the corresponding $Y'$ accordingly. We start off with $X'$. We apply the same set of replacements with only two exceptions: Suppose we want to replace the sample in the slot $\ell$ with $k$ according to the original set of replacement, then we see if $k$ is not in $R'$, we carry on the replacement with $k' = R' \setminus k$. Also, if the $\ell$ is not in $S'$, we will choose slot $\ell' = S' \setminus S$, pick the slot $\ell'$ for the replacement. After performing all the replacement we get $Y'$ which has Hamming distance at most four to $Y$. It is not hard to see that we can map $Y'$ to $Y$ similarly, so there exists a matching between the $Y$'s, and the $Y'$'s. We define the coupling $\mathcal{C}$ to be a probability distribution over $\mathcal{X}^* \times \mathcal{X}^*$, where the probability of $(Y, Y')$ according to the above definition is $1/|R|!$, and it is zero for the rest of the pairs.

**Case 2:** $R$ and $S$ have one extra member: $R' = R \cup \{k\}$, and $S' = S \cup \{\ell\}$. Assume there exists a set of replacements that turns $X$ into $Y$. We construct $|R| + 1$ sets of replacements that turn $X'$ into $Y_1', Y_2', \ldots, Y_{|R|+1}'$. We start off with $X'$. We choose one of the replacement in the set which replaces the sample in slot $\ell'$ by $k'$. Then, we perform all the replacements on $X'$ except the one that is left out. Now, we do the following: We replace the sample in slot $\ell$ by $k'$ and the sample in slot $\ell'$ by $k$. Clearly, we found an assignment between $R'$ and $S'$, so we construct $Y_1', \ldots, Y_{|R|}'$. We also perform all the replacement in the set, and in addition to that, we replace the sample in slot $\ell$ by $k$ to obtain $Y_{|R|+1}'$. It is not hard to see that given $Y'$, we can construct $Y$ as well, so there is a matching between $Y$ and the $Y_t'$'s. Also, $Y$ and the $Y'$'s have a Hamming distance of at most three. Now, we define the coupling $\mathcal{C}$.

---

**Algorithm 2** A private procedure for extending the domain

---
 1: **procedure** PRIVATE TESTER$(X, A)$
 2:     $R, S \leftarrow \emptyset$.
 3:     **for** $i = 1, 2, \ldots, n$ **do**
 4:         **if** $r_i(X) \geq 1$ **then**
 5:             $R \leftarrow R \cup \{r_i(X) \text{ copies of } i\}$
 6:         **if** $s_i(X) \geq 1$ **then**
 7:             $S_i \leftarrow$ Set of the smallest $s_i(X)$ indices of the entries in the bottom row of $X$ which contains $i$.
 8:             $S \leftarrow S \cup S_i$.
 9:     $S \leftarrow |R|$ smallest element in $S$.
10:     **for** each $k \in R$ **do**
11:         $\ell \leftarrow$ a random element in $S$.
12:         $S \leftarrow S \setminus \{\ell\}$.
13:         $X - \text{bottom}(\ell) \leftarrow k$.
14:     Output $\mathcal{A}(X)$.

---

We set the probability of the pairs $(Y, Y_t')$ to be $1/(|R| + 1)!$ for $t = 1, \ldots, |R| + 1$. It is clear that each $Y$ appears with probability $(|R| + 1)/(|R| + 1)! = 1/|R|!$. Thus, the desired coupling exists.

Note that in both case, there exists a coupling $\mathcal{C}$ such that each pair drawn from $\mathcal{C}$ have a Hamming distance of at most four. Hence the proof is complete. $\qquad\square$

### C.5  Proving privacy guarantee after extending the domain

As we describe $\mathcal{B}$ at a high level before, now we formally described it in Algorithm 2. Below, we formally show that the algorithm is differentially private as well.

**Lemma C.6.** *Assume $\mathcal{A}$ is a $\xi/4$-differentially private algorithm over $\mathcal{X}^*$ with parameter $A \geq 12 \ln n/\delta'$ that output the correct answer with probability at least $1 - \delta$. Algorithm 2 is a $\xi$-differentially private algorithm over $\mathcal{X}$. which outputs the correct answer with probability at least $1 - \delta - \delta'$.*

**Proof:** First, we claim that the algorithm changes $X$ with probability at most $\delta'$. Assume $s$ is a Poisson random variable with parameter $\lambda$, and let $X$ be the set of $2s$ samples from a distribution $p$. Using Poissonization method, we can think of $t_i(X)$ and $b_i(X)$ as two Poisson random variables with mean $\lambda_i := p(i) \cdot \lambda$. Now, we bound the probability that $t_i(X)/(b_i(X) + 1)$ become larger than one. If $\lambda_i$ is zero, then $t_i(X)$ and $b_i(X)$ must be zero, so the ratio is below $A$. Let $B = b_i(X)/\lambda_i$. We consider the following cases for $\lambda_i$:

Case 1: $\lambda_i \leq A/2$. By the concentration of a Poisson random variables, we have the following:

$$\mathbf{Pr}\left[\frac{t_i(X)}{b_i(X) + 1} \geq A\right] \leq \mathbf{Pr}[t_i(X) - \lambda_i \geq A - \lambda_i] \leq \exp\left(-\frac{(A - \lambda_i)^2}{2A}\right)$$
$$\leq \exp\left(-\frac{A}{8}\right)$$

Case 2: $\lambda_i > A/2$. Clearly, we have:

$$\mathbf{Pr}\left[\frac{t_i(X)}{b_i(X) + 1} \geq A\right] \leq \mathbf{Pr}[t_i(X) \geq A \cdot b_i(X) + A] = \mathbf{Pr}[t_i(X) \geq A \cdot B \cdot \lambda_i + A]$$

Now, if $A \cdot B \geq 2$, we obtain:

$$\mathbf{Pr}\left[\frac{t_i(X)}{b_i(X) + 1} \geq A\right] \leq \mathbf{Pr}[t_i(X) - \lambda_i \geq \lambda_i + A] \leq \exp\left(-\frac{(A + \lambda_i)^2}{2(A + 2\lambda_i)}\right)$$
$$\leq \exp\left(-\frac{\lambda_i^2}{6\lambda_i}\right) \leq \exp\left(-\frac{A}{12}\right)$$

If $A \cdot B = A\,b_i(X)/\lambda_i < 2$, it means that $b_i(X)$ is at most $2\lambda_i/A$. Thus, we have:

$$
\begin{aligned}
\mathbf{Pr}\left[\frac{t_i(X)}{b_i(X)+1} \geq A\right] &\leq \mathbf{Pr}\left[b_i(X) \leq \frac{2\,\lambda_i}{A}\right] = \mathbf{Pr}\left[\lambda_i - b_i(X) \geq \frac{(A-2)\cdot\lambda_i}{A}\right] \\
&\leq \exp\left(-\frac{(A-2)^2\,\lambda_i^2}{2\,A^2\left(\frac{2A-2}{A}\cdot\lambda_i\right)}\right) = \exp\left(-\frac{(A-2)^2}{2\,A\,(2\,A-2)}\cdot\lambda_i\right) \\
&\leq \exp\left(-\frac{\lambda_i}{6}\right) \leq \exp\left(-\frac{A}{12}\right)
\end{aligned}
$$

where the second to last inequality is true when $A \geq 10$.

In all of the cases above, The probability that the ratio associated with element $i$ goes above $A$ is at most $\exp(-A/12) \leq \delta'/n$. By union bound, the probability of having at least one $i$ with the ratio above $A$ is at most $\delta'$. Observe that if all the ratios are below $A$, all the $r_i(X)$'s will be zero. Thus, the algorithm does not change $X$ with probability $1 - \delta'$. Also, if $\mathcal{A}$ outputs the correct answer with probability at least $1 - \delta$, then $\mathcal{B}$ outputs the correct answer with probability at least $1 - \delta - \delta'$.

Now, we show that $\mathcal{B}$ is private. In Lemma C.5, we show our mapping has the following property: Let $X$ and $X'$ in $\mathcal{X}$ be two datasets with Hamming distance at most one. Let $Y$ and $Y'$ be the randomized datasets that $X$ and $X'$ are mapped to. There exists a coupling $\mathcal{C}$ between $Y$ and $Y'$ where the Hamming distance between any $(Y, Y')$ with non-zero probability in $\mathcal{C}$ is at most four. The existence of the coupling and the fact that $\mathcal{A}$ is an $\xi/4$ private algorithm help us to prove the privacy guarantee for $\mathcal{B}$. Let $O$ be an arbitrary output for $\mathcal{B}$. In the context of our paper $O$ can be accept or reject. Below, we show the probability of outputting $O$ on two neighboring dataset $X$ and $X'$ with Hamming distance one, is the same up to a $e^\xi$ factor.

$$
\begin{aligned}
\mathbf{Pr}[\mathcal{B}(X) = O] &= \sum_Y \mathbf{Pr}[\mathcal{A}(Y) = O] \cdot \mathbf{Pr}[X \text{ is mapped to } Y] \\
&= \sum_{Y,Y'} \mathbf{Pr}[\mathcal{A}(Y) = O] \cdot \mathcal{C}(Y, Y') \\
&\leq \sum_{Y,Y'} e^{(\xi/c)\cdot|Y-Y'|}\mathbf{Pr}[\mathcal{A}(Y') = O] \cdot \mathcal{C}(Y, Y') \\
&\leq \sum_{Y,Y'} e^\xi \mathbf{Pr}[\mathcal{A}(Y') = O] \cdot \mathcal{C}(Y, Y') \\
&\leq \sum_{Y'} e^\xi \mathbf{Pr}[\mathcal{A}(Y') = O] \cdot \mathbf{Pr}[X' \text{ is mapped to } Y'] \\
&= e^\xi \mathbf{Pr}[\mathcal{B}(X') = O]
\end{aligned}
$$

Therefore, $\mathcal{B}$ is $\xi$-private on $\mathcal{X}$. $\qquad\square$

# D  Proof of the Lemmas

## D.1  Proof of Lemma 3.1

**Lemma 3.1.** *Suppose* $r$, $b_i$, $s_{i,1}$, $s_{i,2}$, $v_{i,j,1}$, *and* $v_{i,j,2}$ *are quantities defined above. Then, we have:*

$$
\mathbf{E}_r\left[\sum_{j=1}^{b_i}(v_{i,j,1} - v_{i,j,2})^2 - v_{i,j,1} - v_{i,j,2}\,\middle|\,b_i, s_{i,1}, s_{i,2}\right] = \frac{(s_{i,1} - s_{i,2})^2 - s_{i,1} - s_{i,2}}{b_i}\,.
$$

**Proof:** Observe that given $b_i$, $s_{i,1}$, and $s_{i,2}$, the number of instances of element $i$ in each bucket, $v_{i,j,1}$, is random variables drawn from a binomial distribution $\mathbf{Bin}(s_{i,1}, 1/b_i)$. Similarly, $v_{i,j,2}$ is

drawn from $\mathbf{Bin}(s_{i,2}, 1/b_i)$. Thus, we have:

$$\mathbf{E}[v_{i,j,1}] = \frac{s_{i,1}}{b_i}\,, \quad \mathbf{E}[v_{i,j,1}^2] = \mathbf{Var}[v_{i,j,1}] + \mathbf{E}[v_{i,j,1}]^2 = s_{i,1} \cdot \left(1 - \frac{1}{b_i}\right) \cdot \frac{1}{b_i} + \frac{s_{i,1}^2}{b_i^2} = \frac{s_{i,1}}{b_i} + \frac{s_{i,1}^2 - s_{i,1}}{b_i^2}\,,$$

$$\mathbf{E}[v_{i,j,2}] = \frac{s_{i,2}}{b_i}\,, \quad \mathbf{E}[v_{i,j,2}^2] = \mathbf{Var}[v_{i,j,2}] + \mathbf{E}[v_{i,j,2}]^2 = s_{i,2} \cdot \left(1 - \frac{1}{b_i}\right) \cdot \frac{1}{b_i} + \frac{s_{i,2}^2}{b_i^2} = \frac{s_{i,2}}{b_i} + \frac{s_{i,2}^2 - s_{i,2}}{b_i^2}\,.$$

Since $v_{i,j,1}$ is independent from $v_{i,j,2}$, then we have:

$$\mathbf{E}\left[ \sum_{j=1}^{b_i} (v_{i,j,1} - v_{i,j,2})^2 - v_{i,j,1} - v_{i,j,2} \,\middle|\, b_i, s_{i,1}, s_{i,2} \right]$$

$$= \sum_{j=1}^{b_i} \mathbf{E}\left[ (v_{i,j,1} - v_{i,j,2})^2 - v_{i,j,1} - v_{i,j,2} | b_i, s_{i,1}, s_{i,2} \right]$$

$$= b_i \cdot \left( \mathbf{E}\left[ u_{i,1}^2 + v_{i,1}^2 - 2 \cdot v_{i,j,1} \cdot v_{i,j,1} - v_{i,j,1} - v_{i,j,2} | b_i, s_{i,1}, s_{i,2} \right] \right)$$

$$= b_i \cdot \left( \frac{s_{i,1}^2 - s_{i,1}}{b_i^2} + \frac{s_{i,2}^2 - s_{i,2}}{b_i^2} - 2 \cdot \frac{s_{i,1}}{b_i} \cdot \frac{s_{i,2}}{b_i} \right)$$

$$= \frac{(s_{i,1} - s_{i,2})^2 - s_{i,1} - s_{i,2}}{b_i}\,.$$

which completes the proof. $\qquad\square$

**Lemma A.1.** *Assume $F$ is a random set of samples to be used for flattening. Then, we have:*

$$\mathbf{E}_F\left[ d_{\max}^{(F)} \right] \leq \Theta\left( \mathbf{E}_F\left[ d_{\min}^{(F)} \right] + \mathbf{E}_F\left[ \|p^{(F)} - q^{(F)}\|_2^2 \right] \right)$$

**Proof:** Given a random set $F$, we the $\ell_2$-norm of $p$ and $q$ are two random variables: $\|p^{(F)}\|$ and $\|p^{(F)}\|$. Recall that $d_{\max}^{(F)}$ and $d_{\min}^{(F)}$ are the minimum and the maximum of $\|p^{(F)}\|$ and $\|p^{(F)}\|$ respectively. Consider an event, namely $E$, over the randomness of $F$ that indicates $d_{\max}^{(F)}$ is at most $3 \cdot d_{\min}^{(F)}$. Also, let $\overline{E}$ indicate the complimentary event, when $d_{\max}^{(F)}$ is greater than $3 \cdot d_{\min}^{(F)}$. Using Observation D.1, in this latter case, there exists a constant $c$ such that $d_{\max}^{(F)}$ is at most $c \cdot \|p^{(F)} - q^{(F)}\|_2^2$.

Hence, we have:

$$\mathbf{E}_F\left[ d_{\max}^{(F)} \right] = \mathbf{E}_F\left[ d_{\max}^{(F)} \,\middle|\, E \right] \cdot \mathbf{Pr}_F[E] + \mathbf{E}_F\left[ d_{\max}^{(F)} \,\middle|\, \overline{E} \right] \cdot \mathbf{Pr}_F[\overline{E}]$$

$$\leq 3 \cdot \mathbf{E}_F\left[ d_{\min}^{(F)} \,\middle|\, E \right] \cdot \mathbf{Pr}_F[E] + c \cdot \mathbf{E}_F\left[ \|p^{(F)} - q^{(F)}\|_2^2 \,\middle|\, \overline{E} \right] \cdot \mathbf{Pr}_F[\overline{E}]$$

$$\leq 3 \cdot \mathbf{E}_F\left[ d_{\min}^{(F)} \,\middle|\, E \right] \cdot \mathbf{Pr}_F[E] + 3 \cdot \mathbf{E}_F\left[ d_{\min}^{(F)} \,\middle|\, \overline{E} \right] \cdot \mathbf{Pr}_F[\overline{E}]$$

$$+ c \cdot \mathbf{E}_F\left[ \|p^{(F)} - q^{(F)}\|_2^2 \,\middle|\, E \right] \cdot \mathbf{Pr}_F[E] + c \cdot \mathbf{E}_F\left[ \|p^{(F)} - q^{(F)}\|_2^2 \,\middle|\, \overline{E} \right] \cdot \mathbf{Pr}_F[\overline{E}]$$

$$= \Theta\left( \mathbf{E}_F\left[ d_{\min}^{(F)} \right] + \mathbf{E}_F\left[ \|p^{(F)} - q^{(F)}\|_2^2 \right] \right)$$

Therefore, the proof is complete. $\qquad\square$

**Observation D.1.** *If $\|p\|_2^2 \geq C \cdot \|q\|_2^2$ for a constant $C > 1$, then $\|p - q\|_2^2 = \Theta(\|p\|_2^2)$.*

**Proof:** By the Cauchy-Schwarz inequality, we have:

$$\left(\sum_i (p_i - q_i)^2\right) \cdot \left(\sum_i (p_i + q_i)^2\right) \geq \left(\sum_i (p_i - q_i)(p_i + q_i)\right)^2 \quad \Rightarrow$$

$$\|p - q\|_2^2 \cdot \left(2 + \frac{2}{C}\right) \cdot \|p\|_2^2 \geq \|p - q\|_2^2 \cdot \left(\sum_i 2(p_i^2 + q_i^2)\right) \geq \|p - q\|_2^2 \cdot \left(\sum_i (p_i + q_i)^2\right)^2 \geq \left(\sum_i p_i^2 - q_i^2\right)^2 \quad \Rightarrow$$

$$\|p - q\|_2^2 \cdot \left(2 + \frac{2}{C}\right) \cdot \|p\|_2^2 \geq \left(\sum_i p_i^2 - q_i^2\right)^2 \geq \left(1 - \frac{1}{C}\right)^2 \cdot \|p\|_2^4. \quad \Rightarrow$$

$$\|p - q\|_2^2 \geq \left(\frac{(1 - 1/C)^2}{2 + 2/C}\right) \cdot \|p\|_2^2 = \Omega(\|p\|_2^2).$$

On the other hand, we have:

$$\|p - q\|_2^2 = \sum_i (p_i - q_i)^2 \leq \sum_i p_i^2 + q_i^2 \leq \left(1 + \frac{1}{C}\right) \cdot \|p\|_2^2 = O\left(\|p\|_2^2\right).$$

$\square$

**Lemma B.3.** *Assume random variable $x$ is drawn from $\mathbf{Poi}(\lambda)$. If $\lambda$ is at least $1.5 \cdot \ln(1/c)$, then the probability of $x$ being larger than $3\lambda$ is at most $1 - c$.*

**Proof:** We use the tail bound for the Poisson distribution we have:

$$\mathbf{Pr}_x[x \geq \lambda + 2\lambda] \leq \exp\left(-\frac{(2\lambda)^2}{2 \cdot (2 + 1) \cdot \lambda}\right) \leq \exp(-2\lambda/3) \leq c.$$

Thus, the proof is complete. $\square$

**Lemma B.4.** *Assume we have $n$ independent random variables $x_1, x_2, \ldots, x_n$ in the range $[0, +\infty)$. Suppose each $x_i$ is at least $A_i$ with probability $p \geq 0.95$ where $A_i$ is a fixed number. Then, with probability at least $0.9$, $\sum_{i=1}^n x_i$ is at least $0.1 \sum_{i=1}^n A_i$.*

**Proof:** We define another set of random variables, $y_i$'s, as follows:

$$y_i = \begin{cases} A_i & \text{with probability p} \\ 0 & \text{with probability 1-p} \end{cases}$$

Clearly, the expected value of $\sum_{i=1}^n y_i$ is $p \cdot \sum_{i=1}^n A_i$. Note that we can see $y_i$ as $A_i$ multiplied by a Bernoulli random variable with bias $p$. Thus, the variance of $\sum_{i=1}^n y_i$ is:

$$\mathbf{Var}\left[\sum_{i=1}^n y_i\right] = \sum_{i=1}^n \mathbf{Var}[y_i] = \sum_{i=1}^n A_i^2 \mathbf{Var}[\mathbf{Ber}(p)] = p(1 - p) \cdot \sum_{i=1}^n A_i^2.$$

Now, by the Chebyshev inequality, we can bound the probability of being far from their expectation:

$$\mathbf{Pr}\left[\sum_{i=1}^n y_i \leq 0.1 \cdot \mathbf{E}\left[\sum_{i=1}^n y_i\right]\right] \leq \mathbf{Pr}\left[\left|\sum_{i=1}^n y_i - \mathbf{E}\left[\sum_{i=1}^n y_i\right]\right| \geq 0.9 \cdot \mathbf{E}\left[\sum_{i=1}^n y_i\right]\right]$$

$$\leq \frac{\mathbf{Var}[\sum_{i=1}^n y_i]}{0.9^2 \cdot \mathbf{E}[\sum_{i=1}^n y_i]^2} \leq \frac{p(1 - p) \sum_{i=1}^n A_i^2}{0.9^2 \, p^2 \cdot (\sum_{i=1}^n A_i)^2} \leq \frac{p(1 - p)}{0.9^2 \, p^2} \leq 0.1.$$

Observe that $y_i$'s are defined such that the probability of $\sum_{i=1}^n x_i > a$ for any number $a$ is at least the probability of $\sum_{i=1}^n y_i > a$. Thus, we have:

$$\mathbf{Pr}\left[\sum_{i=1}^n x_i \geq 0.1 \sum_{i=1}^n A_i\right] \geq \mathbf{Pr}\left[\sum_{i=1}^n y_i \geq 0.1 \sum_{i=1}^n A_i\right] \geq 0.9.$$

Hence, the proof is complete. $\square$

**Lemma C.4.** *Let $x_1, x_2, \ldots, x_n$ be $n$ non-negative random variables. Suppose there exist two constants $c$ and $p$, both at most one, such that for each random variable $x_i$, we have:*

$$\mathbf{Pr}[\, x_i < c \cdot \mathbf{E}[\, x_i]\,] \leq p\,,$$

*Then, one can show:*

$$\mathbf{Pr}\left[\sum_{i=1}^{n} x_i < \frac{c \cdot \sum_{i=1}^{n} \mathbf{E}[\, x_i]}{10}\right] \leq \frac{10\,p}{9}\,.$$

**Proof:** At a high level, we expect each random variable $x_i$ to "contributes" to the sum of $x_i$'s by $\mathbf{E}[\, x_i]$. If a random variable $x_i$ is at least $c\,\mathbf{E}[\, x_i]$, it is contributing "enough" to the sum. While otherwise, the sum "misses" a contribution of amount $\mathbf{E}[\, x_i]$. The main idea is to show that total amount that the sum misses is not too large.

More Formally, for each $i$, we define an auxiliary random variables $y_i$ as below. Roughly speaking $y_i$ indicates how much the sum is missing due to a low $x_i$:

$$y_i = \begin{cases} \mathbf{E}[\, x_i] & \text{if } x_i < c \cdot \mathbf{E}[\, x_i] \\ 0 & \text{otherwise} \end{cases}$$

First, we claim that the sum of $y_i$'s is not too large since we have:

$$\mathbf{E}\left[\sum_{i=1}^{n} y_i\right] = \sum_{i=1}^{n} \mathbf{E}[\, x_i] \cdot \mathbf{Pr}[\, x_i < c \cdot \mathbf{E}[\, x_i]] \leq p \cdot \sum_{i=1}^{n} \mathbf{E}[\, x_i]\,.$$

Using Markov's inequality, the sum of $y_i$'s cannot be larger than $0.9 \cdot \sum_{i=1}^{n} \mathbf{E}[\, x_i]$ with probability more than $10\,p/9$. Hence, with probability $1 - 10\,p/9$, we may assume $\sum_{i=1}^{n} y_i$ is at most $0.9 \cdot \sum_{i=1}^{n} \mathbf{E}[\, x_i]$.

Now, we show that the sum of $x_i$'s cannot be too small when the sum of $y_i$'s is less than $0.9 \cdot \sum_{i=1}^{n} \mathbf{E}[\, x_i]$. Too see this, let $I$ be the set of indices $i$ for which $x_i \geq c \cdot \mathbf{E}[\, x_i]$. Then, one can obtain:

$$\sum_{i=1}^{n} x_i \geq \sum_{i \in I} x_i \geq c \cdot \sum_{i \in I} \mathbf{E}[\, x_i] = c \cdot \left(\sum_{i=1}^{n} \mathbf{E}[\, x_i] - \sum_{i \notin I} \mathbf{E}[\, x_i]\right)$$

$$= c \cdot \left(\sum_{i=1}^{n} \mathbf{E}[\, x_i] - \sum_{i \notin I} y_i\right) = c \cdot \left(\sum_{i=1}^{n} \mathbf{E}[\, x_i] - \sum_{i=1}^{n} y_i\right)$$

$$\geq \frac{c \cdot \sum_{i=1}^{n} \mathbf{E}[\, x_i]}{10}\,,$$

which concludes the lemma. $\qquad\square$

**Lemma D.2.** *Assume $x$ is binomial random variable with $n$ trials and bias $p$. Then, the following is true.*

$$\mathbf{E}_x\left[\frac{1}{x+1}\right] \leq \min\left(\frac{1}{p \cdot (n+1)}, 1\right)$$

**Proof:**

$$\mathbf{E}_x\left[\frac{1}{x+1}\right] = \frac{1}{p \cdot (n+1)} \sum_{x=0}^{n} \frac{n+1}{x+1} \binom{n}{x} p^{x+1}(1-p)^{n-x}$$

$$= \frac{1}{p \cdot (n+1)} \sum_{y=1}^{n} \binom{n+1}{y} p^y (1-p)^{(n+1)-y} = \frac{1 - (1-p)^{n+1}}{p \cdot (n+1)}$$

$$\leq \min\left(\frac{1}{p \cdot (n+1)}, 1\right)$$

$$\square$$

**Lemma D.3.** *Assume $x$ is binomial random variable with $n$ trials and bias $p$. Then, the following is true.*

$$\mathbf{E}_x\left[\frac{1}{(x+2)(x+1)}\right] \le \min\left(\frac{1}{p^2 \cdot (n+1)(n+2)}, 1\right)$$

**Proof:**

$$\mathbf{E}_x\left[\frac{1}{(x+2)(x+1)}\right] = \frac{1}{p^2 \cdot (n+1)(n+2)} \sum_{x=0}^{n} \frac{(n+2)(n+1)}{(x+2)(x+1)} \binom{n}{x} p^{x+2}(1-p)^{n-x}$$

$$= \frac{1}{p^2 \cdot (n+1)(n+2)} \sum_{y=2}^{n} \binom{n+2}{y} p^y (1-p)^{(n+1)-y}$$

$$= \frac{1 - (1-p)^{n+2} - (n+2)\,p(1-p)^{n+1}}{p^2 \cdot (n+1)(n+2)}$$

$$\le \min\left(\frac{1}{p^2 \cdot (n+1)(n+2)}, 1\right)$$

$\square$

**Lemma D.4.** *Suppose we have a bin with $m$ balls where exactly $t$ of them are red. We draw balls from the bin without replacement. Let $X$ be the number of red balls in the first $s$ trials and let $Y$ be the number of red balls in the next $k$ trials. Then, we have:*

$$\mathbf{E}\left[\frac{X^2}{Y+1}\right] \le \min\left(\frac{2\,(s-1)t}{(k+1)}, 2\,t^2\right) + t\,.$$

**Proof:** We write the expectation explicitly:

$$\mathbf{E}\left[\frac{X^2}{Y+1}\right] \le \sum_a \sum_b \frac{a^2}{b+1} \cdot \mathbf{Pr}[X=a] \cdot \mathbf{Pr}[Y=b] = \sum_a \sum_b \frac{a^2}{b+1} \cdot \frac{\binom{s}{a}\binom{k}{b}\binom{m-s-k}{t-a-b}}{\binom{m}{t}}$$

$$= \sum_{a\ge 2} \sum_b \frac{2\,a\,(a-1)}{b+1} \frac{\binom{s}{a}\binom{k}{b}\binom{m-s-k}{t-a-b}}{\binom{m}{t}} + s \cdot \sum_b \frac{1}{b+1} \frac{\binom{k}{b}\binom{m-s-k}{t-1-b}}{\binom{m}{t}}$$

$$= \sum_{a\ge 2} \sum_b \frac{2\,a\,(a-1)}{b+1} \frac{\binom{s}{a}\binom{k}{b}\binom{m-s-k}{t-a-b}}{\binom{m}{t}} + s \cdot \sum_b \frac{1}{b+1} \frac{\binom{k}{b}\binom{m-s-k}{t-1-b}}{\binom{m}{t}}$$

$$= \frac{2\,s(s-1)\,t}{(k+1)\,m} \sum_{a\ge 2} \sum_b \frac{\binom{s-2}{a-2}\binom{k+1}{b+1}\binom{m-s-k}{t-a-b}}{\binom{m-1}{t-1}} + \frac{s}{k+1} \cdot \sum_b \frac{\binom{k+1}{b+1}\binom{m-s-k}{t-1-b}}{\binom{m}{t}}$$

We define the two sums in the last line as $A$ and $B$:

$$A := \sum_{a\ge 2} \sum_b \frac{\binom{s-2}{a-2}\binom{k+1}{b+1}\binom{m-s-k}{t-a-b}}{\binom{m-1}{t-1}}, \qquad B := \sum_b \frac{\binom{k+1}{b+1}\binom{m-s-k}{t-1-b}}{\binom{m}{t}}\,.$$

We claim $A$ and $B$ are two probabilities of the following randomized processes, so we can bound them. Suppose we have an urn with $m-1$ balls, $t-1$ of them are red. $A$ is the probability that we get at least one red ball if we draw $k+1$ balls from the bin without replacement. Let $Z$ be the number of red balls we draw after $k+1$ draws. Using Markov's inequality, we get:

$$A = \mathbf{Pr}[Z \ge 1] \le \min(1, \mathbf{E}[Z]) \le \min\left(1, \frac{(t-1)\cdot(k+1)}{(m-1)}\right)$$

Furthermore, we can define $B$ as the following probability: Assume we have an urn of $m$ balls including $t$ red balls. If we draw $(s-1)+(k+1)$ balls from the urn without replacement. $B$ is the probability that non of the $s-1$ draws are red, and there is at least one red draw in the next $k+1$ draws. This is clearly smaller than the probability of seeing at least one red ball in the $k+1$ draws. Thus, similar to the above, we have:

$$B \le \min\left(1, \frac{t\cdot(k+1)}{m}\right)\,.$$

Now, putting all these together, and using the fact that $s \leq m$, we obtain:

$$\mathbf{E}\left[\frac{X^2}{Y+1}\right] \leq \min\left(\frac{2(s-1)t}{(k+1)}, 2t^2\right) + t.$$

946 $\hfill \square$

**Lemma D.5.** *Assume $X$ is a random variable drawn from* $\mathbf{HG}(m,t,k)$*, then*

$$\mathbf{E}_X\left[\frac{1}{(X+1)}\right] \leq \min\left(1, \frac{(m+1)}{(t+1)(k+1)}\right).$$

947 **Proof:** Clearly, the expectation cannot be larger than one since $X \geq 0$. For the other term, by the
948 definition, we can achieve the following bound:

$$\mathbf{E}_x\left[\frac{1}{x+1}\right] = \sum_{x=\max(0,k-(m-t))}^{\min(t,k)} \mathbf{HG}(x;m,t,k) \cdot \frac{1}{x+1} = \sum_x \frac{\binom{t}{x}\binom{m-t}{k-x}}{\binom{m}{k}} \cdot \frac{1}{x+1}$$

$$= \sum_x \frac{m+1}{(t+1)(k+1)} \frac{\frac{t+1}{x+1}\binom{t}{x}\binom{m-t}{k-x}}{\frac{m+1}{k+1}\binom{m}{k}} = \sum_x \frac{m+1}{(t+1)(k+1)} \cdot \frac{\binom{t+1}{x+1}\binom{(m+1)-(t+1)}{(k+1)-(x+1)}}{\binom{m+1}{k+1}}$$

$$\leq \frac{m+1}{(t+1)(k+1)} \cdot \sum_x \mathbf{HG}(x+1;m+1,t+1,k+1) \leq \frac{m+1}{(t+1)(k+1)}$$

949 where the last line is true because the sum of the probabilities according to a distribution is at most
950 one. $\hfill \square$

**Lemma D.6.** *Assume $X$ is a random variable drawn from* $\mathbf{HG}(m,t,k)$*, then*

$$\mathbf{E}_X\left[\frac{1}{(X+2)(X+1)}\right] \leq \min\left(1, \frac{(m+2)(m+1)}{(t+2)(t+1)(k+2)(k+1)}\right).$$

951 **Proof:** Clearly, the expectation cannot be larger than one since $X \geq 0$. For the other term, by the
952 definition, we can achieve the following bound:

$$\mathbf{E}_X\left[\frac{1}{(X+2)(X+1)}\right] = \sum_X \mathbf{HG}(X;m,t,k) \cdot \frac{1}{(X+2)(X+1)}$$

$$= \sum_X \frac{\binom{t}{X}\binom{m-t}{k-X}}{\binom{m}{k}} \cdot \frac{1}{(X+2)(X+1)}$$

$$= \frac{(m+2)(m+1)}{(t+2)(t+1)(k+2)(k+1)} \cdot \sum_X \frac{\binom{t+2}{x+2}\binom{(m+2)-(t+2)}{(k+2)-(x+2)}}{\binom{m+2}{k+2}}$$

$$= \frac{(m+2)(m+1)}{(t+2)(t+1)(k+2)(k+1)} \sum_x \mathbf{HG}(x;m+2,t+2k+2)$$

$$\leq \frac{(m+2)(m+1)}{(t+2)(t+1)(k+2)(k+1)}$$

953 where the last line is true, because the sum of the probabilities in a distribution is at most one. $\hfill \square$

## Footnotes

[3]Needless to say, we are not optimizing constants here.