[Reviews · NeurIPS 2019]

Reviewer 1



Update after reading the author response =============================== The feedback helped me to appreciate the contribution. I changed my scores accordingly. The paper considers the challenge of private distribution testing. Previous results considered "testing samples" approach. This paper considers the flattening samples approach which is more challenging in terms of privacy preserving. To overcome this difficulty, the authors suggest a method that look at all possible permutations of the samples in order to eliminate the sensitivity caused by the flattening process. Overall the suggested algorithm gives an adequate solution for DP in the context of distribution testing. It took a lot of effort from my side to understand the contribution of the paper. The presentation goes back and forth between preliminaries and novel parts, and also between various settings of the problem. I wish that the author presented the setting and the main results in a more concise manner.

Reviewer 2



Additional Comments after feedback period: I've changed my score. I also edited my comment on the typo in line 154-155. ---------------------------------------------------- The paper has a number of typos and should be proofread carefully. Some of them are described below. Overall, though, the paper is written clearly. Since I do not know the literature in differential privacy, it is difficult for me to asesss the originality of the work. Some of the typos in the paper: Line 25: "the 'far'" should be "they 'far'" line 83: "and output" should be "and outputs" line 93 missing right paren line 101: missing citation (?) line 120: typos in definition of Lap(x:gamma) line 128: S and S should be S and S' line 154-155: doesn't parse, needs to be corrected. In the expression in line 154, in the theta notation, the quantity max(||p||_2, ||q||_2) should not be in the denominator. The expression in the footnote also needs to be fixed.

Reviewer 3



Written clearly. Perhaps would benefit from some empirical examples demonstrating the difference in error rate for a two-sample test, as a result of privatizing the test statistic. Comments on notation: Line 120: Laplace distribution has an extra e? Line 157: The reference in [25] indicates that "To begin with, we note that it suffices that only one of ||p||2 and ||q||2 is small. This is essentially because if there is a large difference between the two, this is easy to detect." ... If both are small, might that still make it hard to detect?

[Author Response · NeurIPS 2019]

1. We thank the reviewers for their thoughtful feedback. We will make sure to incorporate all minor editorial recom-
2. mendations in the next revision of our paper. Below we explain our results at a high level and answer the reviewers'
3. questions. We consider two fundamental problems in statistical hypothesis testing: testing independence of a bivariate
4. discrete distribution, and testing closeness (also known as equivalence or two-sample testing) of two unknown discrete
5. distributions given access to unequal sized sample sets from each. We designed minimax sample optimal algorithms
6. (up to a logarithmic factor) for these two problems that satisfy differential privacy. Our main technical contribution is a
7. methodology to privatize the "closeness tester" of [25] that relies on the idea of *flattening* the underlying distributions.
8. (Please see Preliminaries Line 147-173 for a detailed description of this tester).

9. Testing closeness via the flattening technique has played a central role in testing properties of discrete distributions in the
10. non-private setting. Several other distribution properties can be tested via a reduction or direct use of this closeness tester.
11. Examples include identity testing (goodness-of-fit), closeness testing between two unknown distributions with unequal
12. sample sizes, independence testing (in two or higher dimensions), closeness testing for collections of distributions,
13. and testing histograms. For most of these properties, the only known methodology to obtain minimax sample-optimal
14. testers is via a reduction or direct use of the flattening-based closeness tester. This is due in part on the fact that the
15. flattening-based testers naturally allows us exploit the potential structure of the underlying distributions.

16. Despite the importance of the flattening technique in the non-private setting, prior to our work there were no differentially
17. private tester that could make the flattening step private. The main barrier for designing differentially private testers
18. using this method is the unstable nature of the statistic when we use flattening: even changing one sample in the
19. flattening step can drastically change the behavior of the statistic and the tester — whereas a differentially private tester
20. must be stable if a single sample is changed.

21. We give the first differentially private tester that achieves privatizing the flattening-based closeness tester. In particular,
22. we design a general differentially private closeness tester that allows specific reductions for the flattening of the
23. underlying distributions. (See Definition 3.2 for the properties of these specific reductions.) As a corollary, we obtain
24. *the first* minimax sample-optimal and differentially private testers for closeness testing with unequal sized sample sets
25. and independence testing. We circumvent the issue of the unstable statistic by an appropriate *derandomization* of the
26. non-private flattening-based tester: We compute the average statistic over all possible permutations of the samples and
27. carefully analyze its worse-case sensitivity. Furthermore, for independence testing, we provide a novel technique for
28. mapping samples sets with high sensitivity to sample sets with low sensitivity. This technique helps us significantly
29. reduce the sensitivity even further when the worst-case sensitivity is high.

30. **Detecting if $\|p\|_2$ and $\|q\|_2$ are small:** The $\ell_2$-norm of a discrete distribution over $[n]$ is always at least $1/\sqrt{n}$, and
31. we can efficiently estimate the $\ell_2$-norm of any distribution up to a constant factor [8]. In our paper, we do not need to
32. detect whether $\|p\|_2 = \Theta(\|q\|_2)$, as is needed in [25]. We circumvent this detection step entirely by a careful analysis
33. of the statistic and achieve a tester with sample complexity $O\left((n/\epsilon^2)\min(\|p\|_2, \|q\|_2)\right)$. Hence, as long as one of the
34. two distributions has small $\ell_2$-norm (a property guaranteed by flattening), our tester is sample-efficient.

35. **Advantages of statistic $\overline{Z}$:** The main advantage of the statistic $\overline{Z}$ is that it has a low sensitivity. The exact improvement
36. in the sensitivity depends on the flattening procedure and the property being tested. We precisely bound the sensitivity
37. for independence testing and closeness testing with unequal sized sample sets in the respective sections in the Appendix.
38. Please see Section B.2 and Section C.2, where we analyze the sensitivity of $\overline{Z}$.

39. **Dependency on the privacy parameter:** We emphasize that our algorithm is always differentially private regardless
40. of the number of samples. The privacy guarantee follows from the properties of the Laplace mechanism. The sample
41. complexities we obtain are necessary to obtain an accurate tester in a differentially private setting. Moreover, our
42. algorithms have the optimal dependencies on the privacy parameter. Please see the table below for a comparison.

| | Independence Testing | Closeness Testing (with unequal sized sample sets) |
|---|---|---|
| Our Results | $\Omega\left(\frac{n^{2/3}m^{1/3}}{\epsilon^{4/3}} + \frac{\sqrt{mn}}{\epsilon^2} + \frac{\sqrt{mn\log n}}{\epsilon\sqrt{\xi}} + \frac{1}{\epsilon^2\xi}\right)$ | $k_1 = \Omega\left(\frac{n^{2/3}}{\epsilon^{4/3}} + \frac{\sqrt{n}}{\epsilon^2} + \frac{\sqrt{n}}{\epsilon\sqrt{\xi}}\right)$ <br> $s = \Theta\left(\frac{n}{\epsilon^2\sqrt{\min(n,k_1)}} + \frac{\sqrt{n}}{\epsilon^2} + \frac{\sqrt{n}}{\epsilon\sqrt{\xi}} + \frac{1}{\epsilon^2\xi}\right)$ |
| Lower Bounds [4, 25] | $\Omega\left(\frac{n^{2/3}m^{1/3}}{\epsilon^{4/3}} + \frac{\sqrt{mn}}{\epsilon^2} + \frac{\sqrt{mn}}{\epsilon\sqrt{\xi}} + \frac{1}{\epsilon\xi}\right)$ | $s = \Omega\left(\frac{n}{\sqrt{k_1}\epsilon^2} + \frac{\sqrt{n}}{\epsilon^2} + \frac{\sqrt{n}}{\epsilon\sqrt{\xi}} + \frac{1}{\epsilon\xi}\right)$ |

[Meta-Review · NeurIPS 2019]

This is a sound technical paper on the question of testing of distribution with privacy constraints, which shows how tests based on splitting techniques making use of a "flattening sample" (with limited l_2 norm of their corresponding distribution) can preserve privacy. The contribution is solid, timely (cf. privacy) and original.